# Direct observation of cation diffusion driven surface reconstruction at van der Waals gaps

Wenjun Cui[1,2], Weixiao Lin[1,2], Weichao Lu[1,2], Chengshan Liu[1], Zhixiao Gao[3], Hao Ma[3], Wen Zhao[3], Gustaaf Van Tendeloo[2,4], Wenyu Zhao [1] ✉, Qingjie Zhang[1] & Xiahan Sang [1,2] ✉

Weak interlayer van der Waals (vdW) bonding has significant impact on the surface/interface structure, electronic properties, and transport properties of vdW layered materials. Unraveling the complex atomistic dynamics and structural evolution at vdW surfaces is therefore critical for the design and synthesis of the next-generation vdW layered materials. Here, we show that Ge/Bi cation diffusion along the vdW gap in layered $GeBi_2Te_4$ (GBT) can be directly observed using in situ heating scanning transmission electron microscopy (STEM). The cation concentration variation during diffusion was correlated with the local $Te_6$ octahedron distortion based on a quantitative analysis of the atomic column intensity and position in time-elapsed STEM images. The in-plane cation diffusion leads to out-of-plane surface etching through complex structural evolutions involving the formation and propagation of a non-centrosymmetric $GeTe_2$ triple layer surface reconstruction on fresh vdW surfaces, and GBT subsurface reconstruction from a septuple layer to a quintuple layer. Our results provide atomistic insight into the cation diffusion and surface reconstruction in vdW layered materials.

Van der Waals (vdW) layered materials are a group of materials stacked with atomically thin layers bonded by a weak interlayer vdW force. As the interlayer vdW bonding is generally orders of magnitude weaker than the intralayer covalent/ionic bonding, the vdW layered materials are easily exfoliated at the vdW gaps, creating atomically thin 2D vdW materials, e.g. graphene[1,2], boron nitride[3,4], transition metal dichalcogenides (TMD)[5,6]. The unique electronic properties and structural relaxation of 2D vdW materials mainly originate from the breakage of the vdW bonds[7], indicating the importance of the seemingly weak interlayer reaction. The atomistic dynamics at the vdW surfaces also play an important role in applications such as 2D vdW heterostructures[3,8,9], ion conductors[7], lithium ion batteries[10], catalysis[11], thermoelectric materials[12], and topological materials[13,14]. Designing novel functional vdW materials critically depends on the understanding of the structural reconstructions and

diffusion mechanism at the vdW surfaces in both 2D and bulk vdW layered materials.

In situ scanning transmission electron microscopy (STEM) has been widely employed to study atomic-scale structural evolutions under thermal field[15–17], electrical field[18,19], and external stimuli such as gas[20,21] and liquid environment[22]. Besides high spatial and temporal resolution, in situ heating STEM avoids the deleterious influence of surface contamination and oxidation. This is performed by investigating pristine surfaces and edges, formed after etching the material using a combination of thermal energy and electron irradiation. For example, complex 2D edge reconstructions and structural evolution, induced by strong intralayer covalent/ionic dangling bonds, have been directly observed for graphene[23], boron nitride[24], TMD[25] and bismuth/antimony layered chalcogenides[14]. Novel electronic[6,26], photonics[3], catalysis[11,27], sensors[28], and magnetic[29] properties have been revealed at

[1]State Key Laboratory of Advanced Technology for Materials Synthesis and Processing, Wuhan University of Technology, Wuhan 430070, China. [2]Nanostructure Research Center, Wuhan University of Technology, Wuhan 430070, China. [3]School of Materials Science and Engineering, China University of Petroleum (East China), Qingdao 266580 Shandong, China. [4]Electron Microscopy for Materials Science (EMAT), University of Antwerp, Antwerp B-2020, Belgium. ✉e-mail: wyzhao@whut.edu.cn; xhsang@whut.edu.cn

the reconstructed edges. However, vdW surface reconstructions that are dominated by weak vdW forces and its impact on the properties are scarcely investigated.

Bismuth/antimony layered chalcogenides such as $Bi_2Te_3$, $Sb_2Se_3$ and their derivatives such as $GeBi_2Te_4$ (GBT) or $Mn(Sn/Pb/Fe)Bi_2Te_4$ are important vdW layered materials for thermoelectric and topological applications[13,30,31]. We investigated the atomistic dynamics governing cation diffusion, surface reconstruction, and surface etching in the vdW layered material GBT using in situ heating experiments via spherical aberration corrected STEM. The abundant defects in GBT such as antisite point defects and misaligned vdW gaps induced by thermal strain, increases the accessibility of the vdW gaps to cations, making GBT the perfect candidate for the direct observation of Ge/Bi cation diffusion along the vdW gaps. The local cation concentration variation and local $Te_6$ octahedron distortion during cation diffusion have been observed and quantitatively analyzed using the atomic column intensity and position in time-elapsed STEM images. A non-centrosymmetric $GeTe_2$ triple layer (TL) surface reconstruction has been observed on the fresh vdW surface, the {0001} crystallographic surfaces, at the edges of pores created by beam irradiation and heating. The out-of-plane etching process of such surfaces is synergistically modulated by in-plane cation diffusion and the induced $GeTe_2$ TL propagation. These results provide atomistic insight into the cation diffusion and surface reconstruction in vdW layered materials.

## Results and discussion

The GBT sample was synthesized by vacuum-melting Ge, Bi and Te powder at 1073 K for 10 h. X-ray diffraction (XRD) confirms the rhombohedral GBT phase with space group $R\bar{3}m$ (Supplementary Fig. 1). The structural unit of GBT is a septuple layer (SL) consisting of seven sublayers Te1-Bi-Te2-Ge-Te2-Bi-Te1 that are covalently bonded (Fig. 1a). The adjacent SLs are separated by vdW gaps (Fig. 1a), where the two terminating Te1 sublayers are bonded by a vdW force. To better describe the structural evolution during cation diffusion and surface reconstruction, the GBT structure can be viewed as a stacking of $Te_6$ octahedrons: the SL structure is built by three stacks of slightly distorted Ge/Bi centered $Te_6$ octahedrons, while the vdW gap consists of heavily compressed unoccupied $Te_6$ octahedrons (Fig. 1b). Two major geometric variables of the $Te_6$ octahedrons are the projected acute angle $\alpha$, and the spacing $h$ along the $c$ axis (Fig. 1b), which vary depending on the local chemical environment. For example, the geometric variables of the Bi-centered, Ge-centered, and unoccupied $Te_6$ octahedron in GBT are ($\alpha = 70°$, $h = 380$ pm), ($\alpha = 70°$, $h = 340$ pm), and ($\alpha = 50°$, $h = 270$ pm), respectively (Supplementary Fig. 2 and Supplementary Table 1). Atomic resolution annular dark field (ADF)-STEM images, acquired along the GBT [$2\bar{1}\bar{1}0$] zone axis (Fig. 1b), show the SL structure (white dashed rod) and $Te_4$ parallelograms (combination of green lines and white dots) that are projected from the $Te_6$ octahedrons. It is worth noting that the two Bi sublayers and the central Ge sublayer show a brighter contrast than the Te sublayers (Fig. 1b), contradictory to the general trend that the STEM image intensity is proportional to the atomic number[32]. This discrepancy is solved by applying atomic resolution EDS mapping (Fig. 1c) which confirms a mixture of Bi/Ge atoms at both the Ge sublayer and Bi sublayers. Quantitative STEM intensity analysis using image simulation suggests that the Bi concentration in Ge is roughly 54% (see Supplementary Figs. 3, 4 for more details), leading to abundant $Ge_{Bi}$ and $Bi_{Ge}$ antisite defects[31,33].

In situ heating experiments from room temperature (RT) to 500 °C were performed using DENSsolutions microelectromechanical system chips. Cross-sectional TEM samples were fabricated using focused ion beam (FIB) and then glued on the chips (see Methods for more details). As the temperature increases from RT to 250 °C, bright fringes (white arrows in Fig. 1d top) appear in ADF-STEM images, and were also recorded in an in situ TEM movie (Supplementary Movie 1

and Supplementary Fig. 5). Low-magnification EDS mapping shows that Ge, Bi and Te are uniformly distributed in the bright contrast regions, indicating that segregation of heavy elements did not occur (Supplementary Fig. 6). Atomic resolution ADF-STEM image (Fig. 1d bottom) reveals that the bright fringes are stacking faults where SLs (white rods) are destroyed, while quintuple layers (QLs, cyan rods), triple layers (TLs, yellow rods), and double layers (DLs, green rods) are induced by the breaking and recombination of sublayers. Distortion of sublayers can be seen by following the vdW gap in Fig. 1d from left to the right, where the gap size gradually decreases, while the number of sublayers remain the same (Supplementary Fig. 7). However, Geometric phase analysis (GPA) confirms the local strain around the bright fringes (Supplementary Fig. 8), indicating that the bright contrast should originate from the local strain and local intermixing of cations and anions. Similar structures[7,34,35] have been reported in vdW layered materials after heat treatments[35,36] or under strain[37].

As the heating temperature further increases to 400 °C, nano-sized parallelogram-shaped pores are generated (Fig. 1e). The edges of the pores are newly created surfaces that are mainly parallel to {0001}, {$01\bar{1}\bar{7}$} and {$01\bar{1}\bar{4}$} crystallographic planes (Fig. 1f–h), providing an opportunity to study the reconstruction of an atomically flat surface without surface contamination, oxidation, or mechanical damage. For example, the {0001} surface reconstruction leads to the formation of a TL structure with lower contrast than the bulk region (Fig. 1f, indicated by yellow arrow). The TL atomic columns have similar STEM intensity compared to the Te column intensity in GBT (Fig. 1f, green line scan), suggesting that the TL mainly consists of Te atoms. The same TL structures are also observed at the reconstructed {$01\bar{1}\bar{7}$} surface (Fig. 1g) and the {$01\bar{1}\bar{4}$} surface (Fig. 1h). EDS mapping confirms that the reconstructed surface has relatively high Te content (Supplementary Fig. 9), which agrees well with the proposed $Te_6$ octahedron structure. In all three cases, the surface TL can be interpreted as connecting $Te_6$ octahedrons that completely cover the flat regions of the surface and wrap around the steps (indicated by yellow arrows), showing great flexibility.

To understand the local chemistry of the surface TL, we recall that the $Te_6$ octahedron geometry is closely related to the cation occupancy (Fig. 1b). By fitting the STEM intensity distribution using a Gaussian distribution[38–42], we obtain the projected positions of the Te atom columns in the TL, and an average $\alpha$ (63°) and $h$ (360 pm) value for the $Te_6$ octahedrons. The TL spacing $h$ substantially exceeds the spacing of the unoccupied $Te_6$ octahedrons in GBT (270 pm), indicating that the TL octahedral interstitial sites should be occupied by cations. For three reasons, we speculate that the cations are mainly Ge. First, the contrast near the octahedral centers is faint, and therefore the cations should have a lower atomic number than Te, which excludes a high concentration of heavy Bi cations. Second, the faint contrast is off-center (inset in Fig. 1f–h), and the Ge cations in GeTe with its non-centrosymmetric crystal structure is also off-center[43]. The argument is supported by the good agreement between the experimental STEM images and the simulated STEM images using the GeTe structure (Fig. 1i). Third, the Ge concentration is higher than Bi concentration at the surface as confirmed by EDS mapping (Supplementary Fig. 9). Although preferential occupation of minority cations near the surface in some similar structures has been reported[14,44–46], a TL terminated surface reconstruction has never been observed before.

To understand the atomistic dynamics governing the pore formation and the surface reconstruction, we investigated the diffusion mechanism using in situ STEM imaging. The cation diffusion from the bulk to the surface along the vdW gap has been directly observed using atomic resolution STEM images snapped at different time during in situ heating (Supplementary Movies 2-4). An edge dislocation with the extra half plane consisting of a Ge/Bi sublayer is present on all the frames (Supplementary Movie 2 and Fig. 2a). From left to right, the

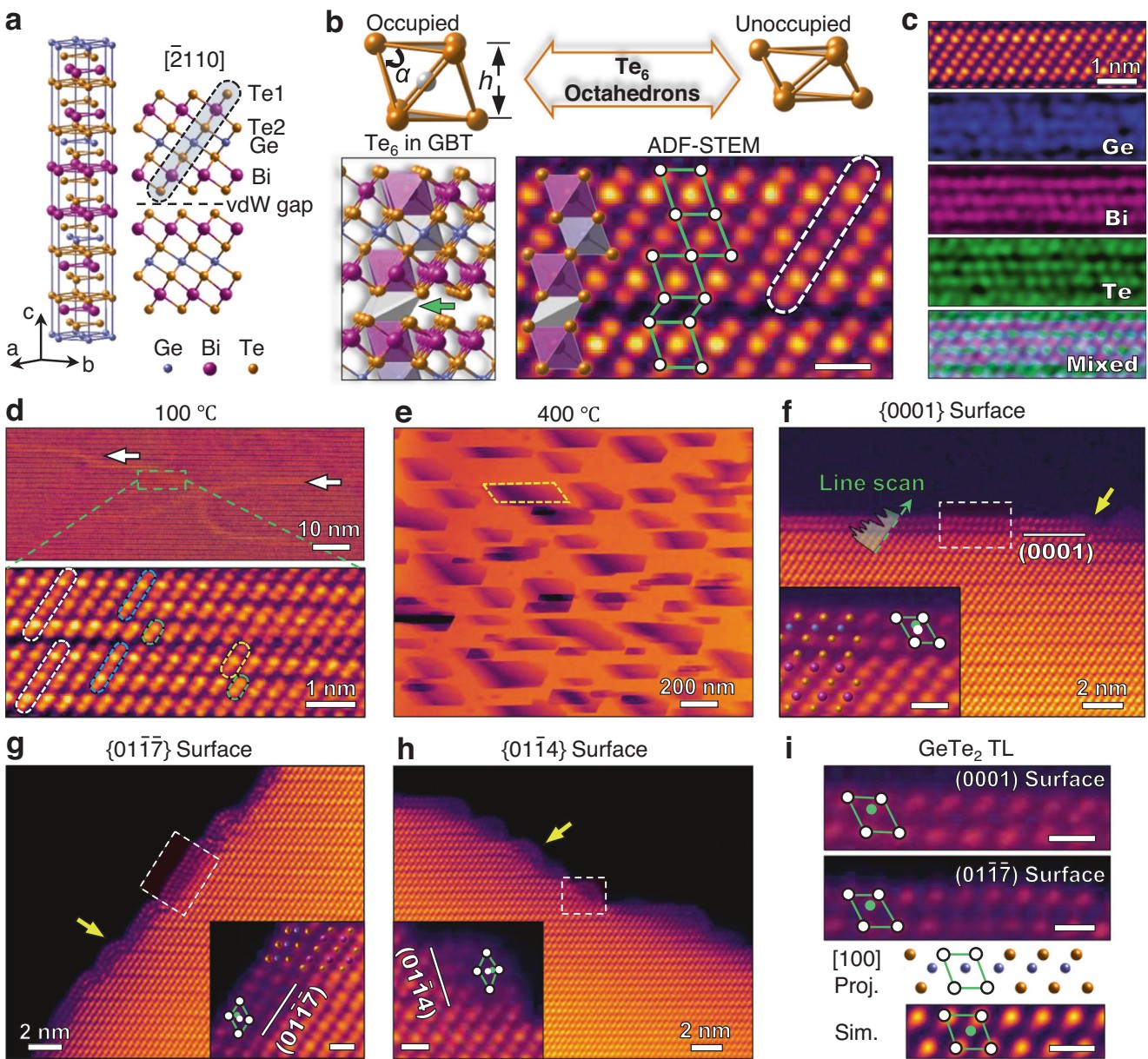

**Fig. 1 | Structure characterization and reconstructed surfaces in GeBi$_2$Te$_4$ (GBT) crystal. a** Crystal structure model of the GBT septuple-layer (SL). **b** Atomic models of occupied (top left) and unoccupied (top right) Te$_6$ octahedrons with geometric parameters $\alpha$ and $h$. The arrangement of the Te$_6$ octahedrons in the GBT crystal structure, and the projected Te$_4$ parallelograms in the ADF-STEM image. **c** Atomic resolution EDS elemental maps of GBT, with Ge (purple), Bi (pink), Te (green) and the mixed. **d** ADF-STEM image of GBT acquired at 100 °C. Septuple layers (SLs), quintuple layers (QLs), triple layers (TLs), double layers (DLs) are indicated by white, cyan, yellow and green dashed rods, respectively. **e** Low magnification ADF-STEM image of nanopores at 400 °C. **f**–**h** Atomic resolution STEM images of (0001), (00$\bar{1}\bar{7}$) and (01$\bar{1}4$) surfaces at 400 °C. The insets are enlarged images from the dashed line boxes. Inset in (**f**) includes DFT optimized interface structure. Te$_6$ octahedrons are marked as Te$_4$ parallelogram (white dots connected by green lines). The geometric centers and the actual cation locations are marked by white and green circles respectively. Experimental and simulated STEM images of surface TL. The scale bars are 0.5 nm in (**b**) and (**i**), and insets in (**f**–**h**).

Ge/Bi sublayer intensity gradually decreases and eventually drops to the background level after passing the dislocation core (cyan arrows in Fig. 2a, b). Between $t = 145\,s$ and $t = 165\,s$, the dislocation core and the extra Ge/Bi sublayer shift to the left by eight Te$_6$ octahedrons (3.03 nm) (Fig. 2b), indicating that some Ge/Bi cations have diffused away, most likely along the vdW gap to the right. Such direct observation of cation diffusion has not been observed before in layered vdW structures. DFT calculation reveals that the energy barriers for Ge (Bi) atoms diffusing along the Bi$_2$Te$_3$ and GBT vdW gaps are 0.53 eV (0.81 eV) and 0.42 eV (0.27 eV), respectively (Supplementary Fig. 10 and Supplementary Table 2). The diffusion energy barriers are comparable with the thermal energy (0.06 eV) at 400 °C.

The Ge/Bi cation diffusion leads to a time-dependent Te$_6$ octahedron distortion around the dislocation core area. The 25 sequentially labeled Te$_6$ octahedrons in the vicinity of the dislocation core are marked in 5 STEM frames acquired from $t = 145\,s$ to $t = 165\,s$ (Fig. 2c). Based on a quantitative analysis of the position and the intensity of the Te atomic columns, the interstitial STEM intensity $I_i$ and geometric parameters ($\alpha$, $h$) of the 25 Te$_6$ octahedrons are summarized in Fig. 2d. The black and red dashed lines represent the supposed values for the Ge/Bi occupied Te$_6$ octahedron and the unoccupied Te$_6$ octahedron at the vdW gap, respectively. The cyan, white and yellow shaded regions represent the initial Ge/Bi extra sublayer region, the transition region, and the vdW gap region at 145 s. For all five frames, the interstitial

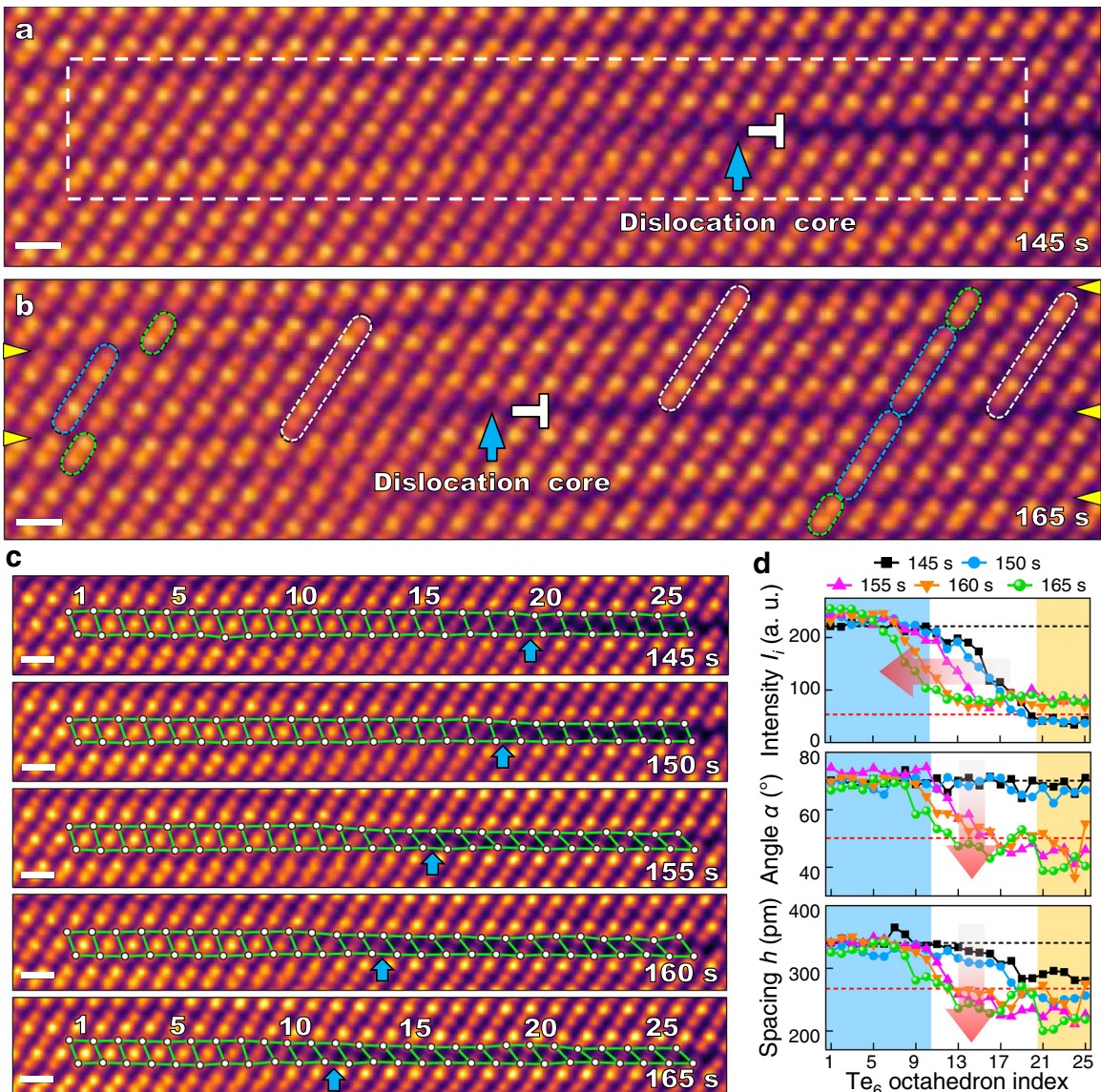

**Fig. 2 | In situ STEM characterization of cations diffusion along a vdW gap.**
**a**, **b** ADF-STEM images acquired at 145 s and 165 s. The dislocation cores are indicated by cyan arrows. SLs, QLs, and DLs structures are indicated by white, cyan and green rods, respectively. VdW gaps are indicated by yellow arrows. **c** A series of aligned ADF-STEM images around the dislocation core area (white dashed box in (**a**)) from 145 s to 165 s. 25 Te$_6$ octahedrons at the dislocation core area are indicated by white circles connected by green lines. The index of the octahedrons runs from 1 to 25 from left to right. **d** Cation atomic column intensity ($I_i$), angle ($\alpha$) and spacing ($h$) of STEM images from 145 s to 165 s as a function of Te$_6$ octahedron index. Blue and yellow shades indicate fully occupied and unoccupied areas at 145 s, respectively. All scale bars are 0.5 nm. Source data are provided as a Source Data file.

STEM intensity $I_i$ starts at the mixed Ge/Bi intensity, and then rapidly decreases to the background level after reaching the dislocation core (cyan arrow in Fig. 2c). The experimental intensity variation can be nicely fitted using the diffusion curves as predicted by Fick's law:

$$I_i = \frac{c_1 + c_0}{2} - \frac{c_1 - c_0}{2}\, erf\left(\frac{x_i - x_s}{l}\right) \qquad (1)$$

where $c_1$ and $c_0$ are intensities at the left and right ends, respectively, *erf* is the error function, $x_s$ is the dislocation core position, and $l$ is the diffusion length defined as:

$$l = 2\sqrt{Dt} \qquad (2)$$

with the diffusivity $D$ and time $t$ (Fig. 3b and Supplementary Fig. 11). The fit results are summarized in Table 1. As time elapses, the

dislocation core position $x_s$ gradually moved to the left (Fig. 2a, b), and the diffusion length $l$ shows a sudden 17.1 % decrease from 1.75 nm at 150 s to 1.45 nm at 155 s. This can be related to the abrupt decrease of $h$ and $\alpha$ back to the supposed vdW gap values at the transition and vdW gap regions (indicated by red arrows in Fig. 2d), causing the shear strain to be released. In situ GPA confirms the disappearance of the shear strain $\varepsilon_{xy}$ from $t = 155\,s$ and out-of-plane tensile strain $\varepsilon_{yy}$ from $t = 160\,s$. (Fig. 3a and Supplementary Fig. 12). The systematic change of shear strain around the dislocation core is unlikely caused by scan distortion that tends to have random effect on the whole scan line, while here the shear strain only exists at the right part of the dislocation core. Therefore, the larger spacing at the vdW gap under tensile strain leads to the initial higher diffusion distance.

The relationship between the local geometric parameters and the local cation intensity of the 25 Te$_6$ octahedrons from the two STEM frames under strain and the three STEM frames after the strain is

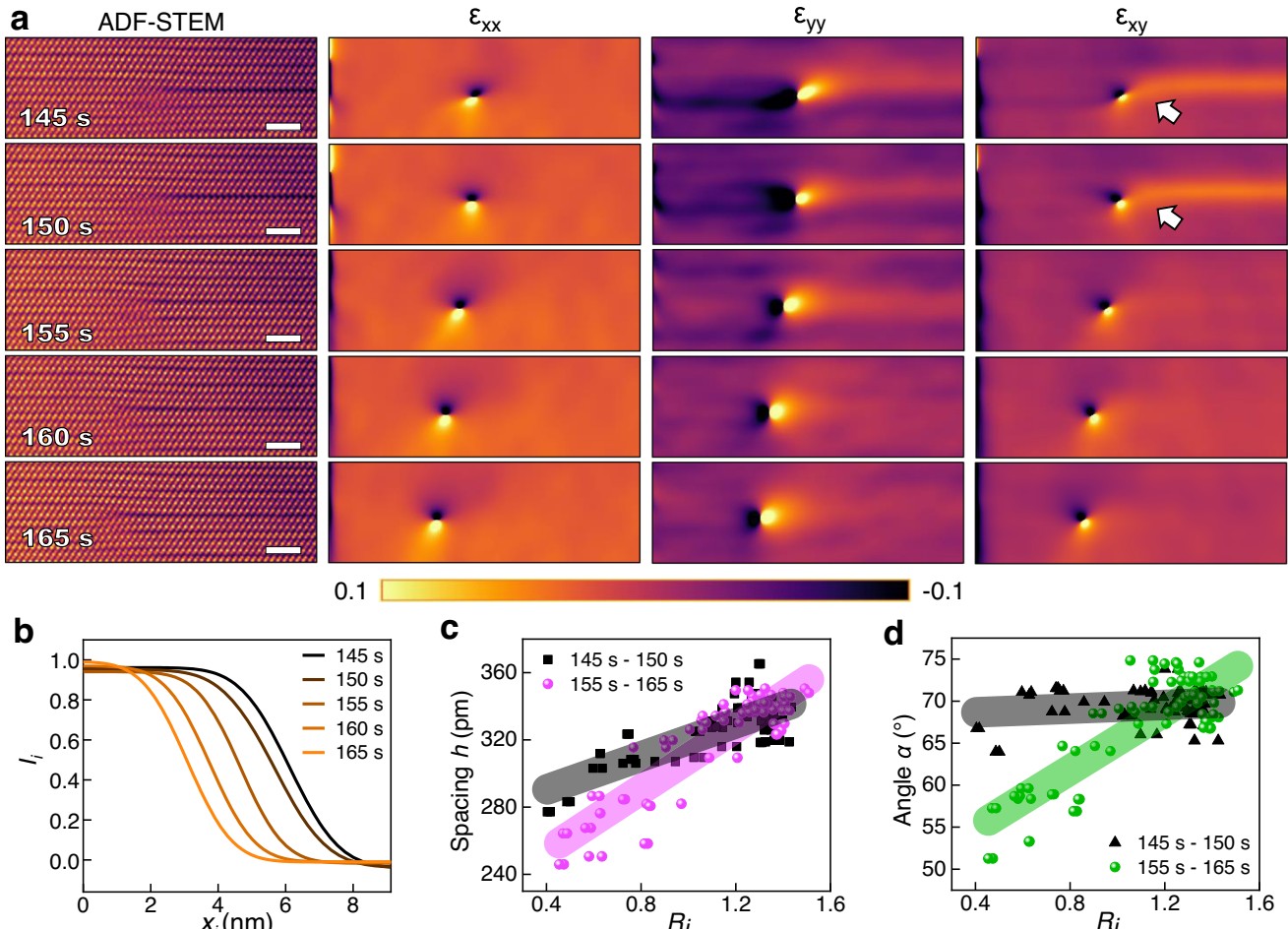

**Fig. 3 | Strain and local distortion during cation diffusion. a** Geometric phase analysis (GPA) of STEM images acquired from 145 s to 165 s. The locations of the shearing strain streak are indicated by white arrows. **b** Fitting curves from different times using Fick's law. **c**, **d** Relationship between local geometric parameters ($h$, $\alpha$) and local cation intensity ($R_i$) before (145–150 s) and after (155–165 s) the strain is relaxed. All scale bars are 2 nm. Source data are provided as a Source Data file.

relaxed is summarized in Fig. 3c, d. Here the normalized cation intensity $R_i$ is defined as $R_i = I_i/I_{Te}$, where $I_{Te}$ is the average intensity of the 4 surrounding Te atomic columns, which can serve as fiducial markers as their intensities are independent of time or structural changes (Supplementary Fig. 13). As $R_i$ increases, the spacing $h$ increases monotonically for all five frames. A linear fitting yields $h = 49.4R_i + 270.6$ with correlation coefficient $r = 0.77$ for $t = 145$ s to 150 s, and $h = 92.3R_i + 216.6$ with correlation coefficient $r = 0.90$ for $t = 155$ s to 165 s, indicating a strong positive correlation between $R_i$ and $h$ with and without the strain (Fig. 3c). On the other hand, the correlation coefficient $r$ between the angle $\alpha$ and $R_i$ is 0.15 and 0.84 for frames before (black triangles) and after (green dots), respectively, indicating that the angle $\alpha$ depends more on the global shear strain than on the local concentration variation (Fig. 3d). DFT calculation confirms that the geometric parameters of $Te_6$ octahedrons critically

depend on cation concentration. (Supplementary Fig. 14). Both the spacing and the angle change abruptly when the cation concentration is around 50%, agreeing well with the sudden relaxation of the shear strain. In situ STEM imaging therefore can potentially be applied to observe cation diffusion such as Li diffusion along vdW gaps in other material systems, with the help of local distortion analysis, and new techniques that can enhance cation contrast and temporal resolution.

Cation diffusion along the vdW gap plays an important role in the $GeTe_2$ TL terminated surface reconstruction. The propagation of the $GeTe_2$ TL at the surface and the formation of new terminated $GeTe_2$ TL driven by cation diffusion were captured using in situ STEM imaging (Supplementary Movie 3 and Fig. 4). At $t = 0$ s, the GBT (0001) surface is terminated by a $GeTe_2$ TL consisting of a top Te sublayer and a bottom Te sublayer, which can be further divided horizontally to Segment 1 (Seg 1) and Segment 2 (Seg 2) by a step indicated by a yellow arrow (Fig. 4a). Seg 1 has clean and clear top and bottom Te sublayers, and is epitaxially attached to the GBT QL (indicated by blue dotted rods) through the vdW gap (Gap 1). For Seg 2 still at the incipient stage, the top Te sublayer is covered by an amorphous cluster, while the bottom Te sublayer is shared with a GBT SL (indicated by white rods). The right part of the Ge/Bi sublayer in this GBT SL is missing, forming an edge dislocation similar to the one observed in Fig. 2. From $t = 0$ s to 155 s, the dislocation core, marked by cyan arrows, and the Ge/Bi extra half plane framed by a green line moves to the left by 9.1 nm (Fig. 4 and Supplementary Fig. 15), which has its origin in the diffusion of the Ge/Bi cations to the right along the newly formed vdW gap (Gap 2).

**Table 1 | Fitting parameters of the intensity curves for different times using Fick's law**

| Time | $x_s$ (nm) | $l$ (nm) |
|------|-----------|----------|
| 145 s | 6.12 | 1.65 |
| 150 s | 5.68 | 1.75 |
| 155 s | 4.66 | 1.45 |
| 160 s | 3.76 | 1.48 |
| 165 s | 3.09 | 1.49 |

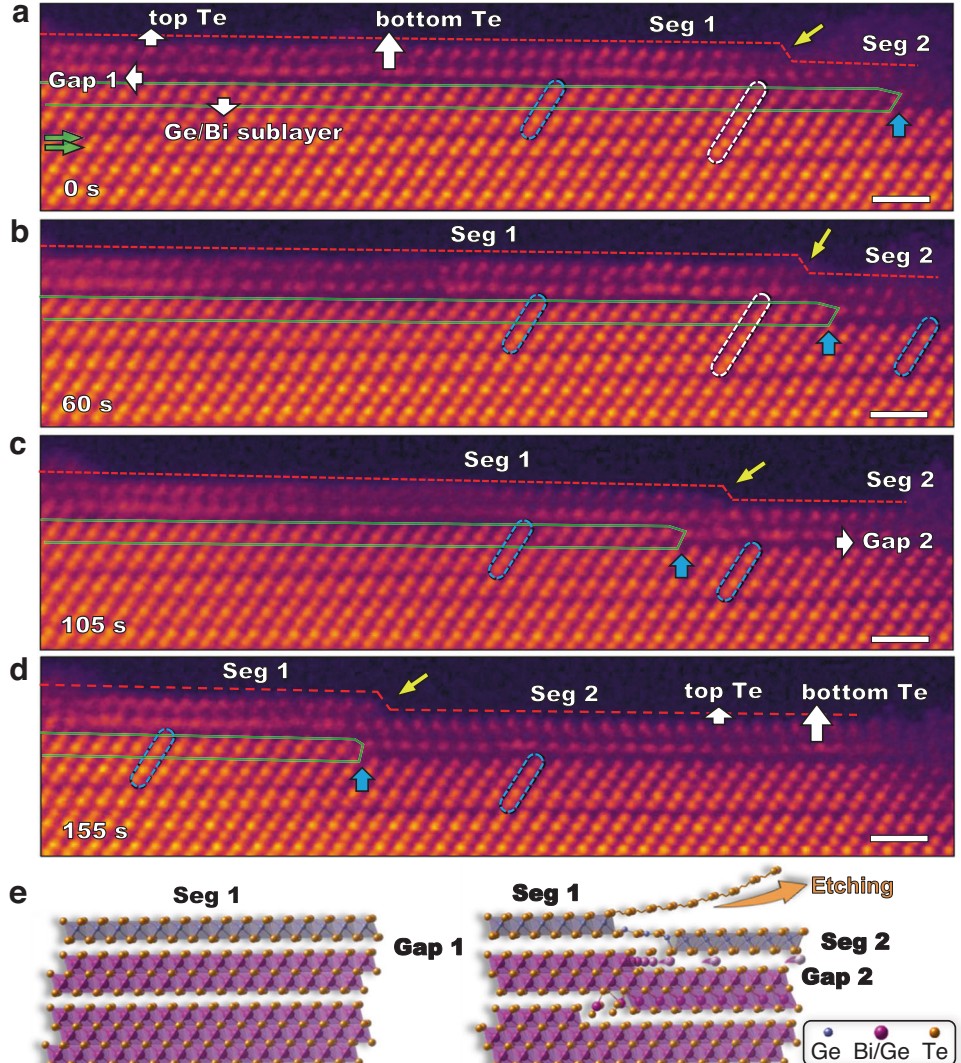

**Fig. 4 | GeTe₂ TL (0001) surface reconstruction.** In situ STEM images of the (0001)-terminated surface acquired at different time: **a** 0 s, **b** 60 s, **c** 105 s and **d** 155 s, at 400 °C. The red dashed line on the surface traces the outline of Seg 1 and Seg 2, and the step between them is indicated by yellow arrows. The extra Ge/Bi half plane is outlined by green solid lines and the dislocation core is indicated by cyan arrows. Top and bottom Te sublayers, Gap 1 and Gap 2 are all pointed by white arrows. **e** Schematic showing of the cation-diffusion mediated surface etching at the (0001) surface, before (left) and after (right). All scale bars are 1 nm.

Concomitantly, Seg 2 propagates out of the amorphous cluster, while Seg 1 and Gap 1 recede to the left (yellow arrows) at the same rate as the expansion of Gap 2 (cyan arrows in Fig. 4b, d). Therefore, the propagation and formation of the GeTe₂ TL on the surface is driven by cation diffusion along the vdW gap. This whole process is schematically shown in Fig. 4e.

During the structural evolution, one striking feature is that the GBT structure adjacent to the GeTe₂ TL always has the QL structure, indicated by blue dotted rods in Fig. 4. As Seg 2 propagates to the left, the underlying GBT QL structure follows its path and expands to the left, while as Seg 1 shrinks, the adjacent GBT QL also recedes at the same pace (Fig. 4b, d). As Seg 1 is gradually replaced by Seg 2, the subsurface QL should have been reduced to a TL. However, the GeTe₂ TL surface reconstruction tends to stabilize a subsurface QL structure. To avoid formation of TL, a DL (indicated by green arrows in Fig. 4) is detached from the SL below and re-attached to the TL to form a new QL. The attachment and detachment of DL is accompanied by inter-mixing of Bi/Ge sublayer and Te sublayer, as demonstrated by the varying $I_{top}/I_{bottom}$ intensity ratio for every atom pair in the DL structure from left to right for different frames (see Supplementary Fig. 15 for more details on the correlation between vdW gap and intensity ratio).

As time evolves, the cations gradually diffuse to the top layer. The vdW gap above the top layer gradually closes, forming the desired QL structure.

The average geometric parameters of the Te₆ octahedrons in the GeTe₂ TL ($\alpha \approx 63°$, $h \approx 360$ pm) (Supplementary Fig. 16) match very well with the experimentally measured parameters for GeTe crystal ($\alpha = 69°$, $h = 360$ pm) (Supplementary Table 1). In situ experiments performed at lower temperature (150 °C, Supplementary Fig. 17) or without prolonged electron beam irradiation (Supplementary Fig. 18) suggest that the observed cation diffusion and GeTe₂ TL surface reconstruction are mainly activated by thermal energy. That means the GeTe₂ TL surface structure could be thermally engineered to tailor functional properties and topological properties. GBT is a topological insulator with bulk energy gap ~ 180 meV. The topological surface state below and above the Dirac point are isolated from the bulk band. It has been reported that the disorder in GBT has little effect on the surface state spin polarization[31]. However, the drastic surface reconstruction shown here changes the surface chemical composition, the stacking number, and possibly the majority point defect and the majority car-rier, which can have an influence on the electric and spin properties. This can further change the relative contribution of topological surface

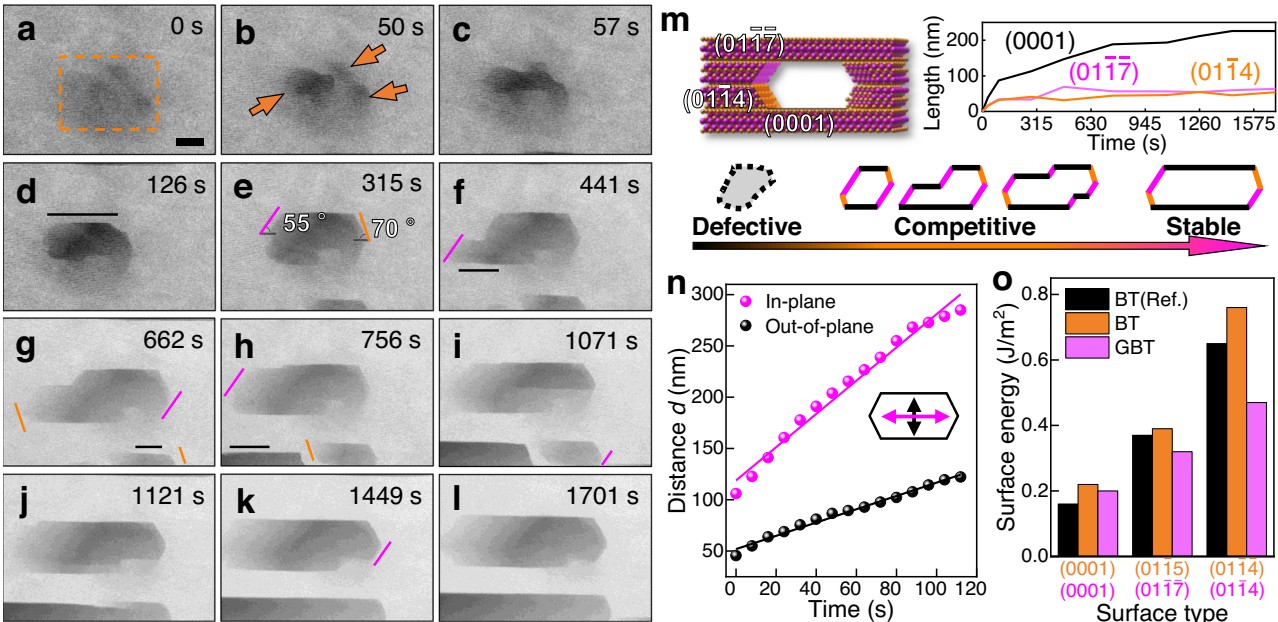

**Fig. 5 | Pore expansion and anisotropic etching. a–l** ADF-STEM images acquired at different times showing the evolution of the pores viewed along the [$\bar{2}$110] zone axis at 400 °C. Black, pink and orange solid lines indicate the newly formed (0001), (01$\bar{1}$7) and (01$\bar{1}$4) surfaces, respectively. The scale bar is 20 nm. **m** Atomic model of a typical pore terminated with three different surfaces (top left). Total projected length of the different surfaces as a function of time (top right). Schematic representation of the typical pore evolution stages (bottom). **n** Average in-plane and out-of-plane distances of the pores as a function of time. **o** Surface energy of the different surfaces in bulk Bi$_2$Te$_3$ and GBT. Source data are provided as a Source Data file.

states to the electron transport. DFT calculation shows that the GeTe$_2$ TL is metallic, while the GeTe bulk material is semiconductor (Supplementary Fig. 19). The TL structure on the surface thus has higher electrical conductivity than the bulk material, which may provide a method to increase electrical conductivity without improving thermal conductivity. The ferroelectric TL structure can spontaneously generate local electrical field, which may introduce extra scattering to the carriers and improve Seebeck coefficient. Therefore, the GeTe$_2$ TL surface reconstruction provides an alternative route to design and build vdW heterostructures for thermoelectric and ferroelectric applications[9].

The surface reconstruction mediated by the expansion of Seg 2 via absorbing Seg 1 results in an out-of-plane etching of the GBT (0001) surface by one Te sublayer, as indicated by the steps (yellow arrows) in Fig. 4a–d. The bottom Te sublayer of Seg 1 is almost aligned with the top Te sublayer of Seg 2, suggesting that the lattice distortions caused by diffusion of the Ge/Bi cations along Gap 2 and the occupation of Ge atoms at Gap 1 seem to cancel each other. Therefore, the out-of-plane etching mainly originates from a peeling of the Seg 1 top Te sublayer to avoid the formation of Seg 1 stacking on Seg 2, i.e., the Ge$_2$Te$_3$ QL structure. More examples of such reconstruction can be seen in Supplementary Figs. 20–22. As predicted by DFT, GeTe$_2$ TL is more stable than Ge$_2$Te$_3$ QL structure for high Te chemical potential (Supplementary Fig. 19a), which matches well with EDS mapping at the surface (Supplementary Fig. 9). The reconstruction images and movie of (01$\bar{1}$7) and (01$\bar{1}$4) surfaces were also recorded in Supplementary Movie 4 and Supplementary Fig. 23. The etching rate of (0001) surface at out-of-plane direction should be significantly lower than other surfaces due to the (0001) surface etching involving long-distance Ge/Bi cation diffusion that is parallel to the surface.

The etching rate of different surfaces was investigated by recording low-magnification STEM images of the pore expansion during in situ heating (Supplementary Movie 5 and Fig. 5a–l). Initially, a dark contrast emerges as vacancies are randomly generated by beam irradiation and thermal agitation (Fig. 5a). The coalescence of

vacancies leads to dark pores with irregular shapes (Fig. 5b), which then slowly evolve into faceted pores (Fig. 5c–e) that are terminated by (0001) (black), (01$\bar{1}$7) (pink), and (01$\bar{1}$4) (orange) crystallographic planes. It has been observed that faceted pores tend to form at thin regions. The expansion of the pores is highly anisotropic, with (0001) surfaces etching much slower than the other surfaces (Fig. 5f–l). As time elapses, the projected length of the (0001) surface increases much faster than other surfaces (Fig. 5m), indicating that the (0001) surface is energetically the most stable. Statistically, the etching rate in the out-of-plane direction ($0.65 \pm 0.02$ nm/s) is three times lower than the in-plane etching ($1.62 \pm 0.06$ nm/s) (Fig. 5n), similar to Bi$_2$Te$_3$[47]. Eventually, the pores turn into narrow hexagons or parallelograms that are mainly terminated by three low energy surfaces (Fig. 5l, m). Thermodynamically, DFT calculations without surface reconstruction reveal that the surface energy ($\gamma$) of Te-terminated {0001}, {01$\bar{1}$7}, {01$\bar{1}$4} surfaces in GBT are 0.20, 0.32, 0.47 J/m$^2$, respectively (Fig. 5o), which show similar trends as the {0001} (0.22), {01$\bar{1}$5} (0.39), {01$\bar{1}$4} (0.76) surfaces in Bi$_2$Te$_3$[48,49]. Therefore, the {0001} surface has the lowest surface energy in both QL and SL structures. Kinetically, as revealed earlier, the out-of-plane etching of GeTe$_2$ TL reconstructed {0001} surfaces are activated by in-plane cation diffusion, which significantly limits the etching rate.

In summary, the cation diffusion mechanism at vdW gaps and its impact on the surface reconstruction and etching in the vdW layered material GeBi$_2$Te$_4$ has been systematically investigated using in situ atomic resolution STEM imaging. The cation diffusion is validated by both the variation of the atomic column intensity, and the local distortion of the projected Te$_6$ octahedrons, providing a convincing approach to study the transport mechanism for similar materials with Te$_6$ octahedron building blocks. Moreover, the in-plane cation diffusion critically determines the kinetics of the formation and propagation of the GeTe$_2$ TL on the vdW surfaces, resulting in a low etching rate of the vdW surfaces, which was also confirmed by in situ STEM. The results provide valuable atomistic insight to understand the complex structural evolution at vdW gaps, which can benefit the

design of new heterogenous vdW 2D materials, topological materials, and energy materials.

## Methods

### Material synthesis and TEM sample preparation

Ge powder (99.99%), Bi powder (99.999%) and Te powder (99.999%) with atomic ratio of 1:2:8 were weighed and mixed in a quartz tube, which was sealed at 0.1 Pa and then placed in muffle furnace. The tube was kept at 1073 K for 10 h, and then cooled to temperature with furnace. A 40 μm × 2 μm × 15 μm GeBi$_2$Te$_4$ (GBT) lamella were fabricated using focused ion beam (FIB, Helios NanoLab G3 UC) in a UHV (<10$^{-6}$ mbar) environment, and then welded on the heating and biasing nano-chips using GIS system in FIB. The nanochip was then loaded on a DENSsolutions double-tilt in situ heating holder.

### XRD and STEM characterizations

X-ray diffraction was performed on a Rigaku SmartLab automated multipurpose R-ray diffractometer using Cu K$_\alpha$. STEM imaging and EDS analysis were performed on an FEI double C$_s$ corrected Titan Themis G2 operated at 300 kV and equipped with an X-FEG electron gun. The convergence semi-angle of the probe was 17.8 mrad, the inner and outer collection angles of the STEM images were 48 and 200 mrad, respectively. The screen current used for ADF-STEM imaging and EDS analysis was 50 pA and 100 pA, respectively. During in situ experiments, the sample was heated up to a certain temperature at a heating rate of 1 °C/s. STEM image simulation was performed using multi-slice algorithm implemented in QSTEM package[50] using the same experimental parameters. The thermal diffuse scattering (TDS) was considered using the frozen phonon method with 10 configurations. The Debye Waller factors of Ge, Bi and Te used for image simulation are 2.21, 2.03 and 1.06 Å$^2$, respectively[51]. GPA strain mapping was calculated using (0006) and (01$\bar{1}$7) diffraction spots implemented in Digital Micrograph plugin GPA-v2.0.gt1[52].

### STEM image processing

All STEM images shown in Figs. 1–4 were first background-corrected by filtering out low-frequency information around the (000) diffraction spot, and then blurred using a 2D gaussian distribution with σ = 1, implemented in the commercial software Velox that comes with Titan Themis microscope. For quantitative analysis, the raw data of STEM images were exported from Velox, denoised using a gaussian filter with σ = 2, and then analyzed using the free package CalAtom[41]. The frames in the supplementary movies were batch-filtered using a Gaussian distribution with σ = 1. The atomic columns were fitted individually using Gaussian distribution. For each atomic column, the pixels containing the atomic column were fitted using a Gaussian distribution as defined below:

$$I(x,y) = I_0 + A \exp \left\{ -\frac{\left[ (x - x_g)\cos\theta - (y - y_g)\sin\theta \right]^2}{2\sigma_x^2} - \frac{\left[ (x - x_g)\sin\theta + (y - y_g)\cos\theta \right]^2}{2\sigma_y^2} \right\}$$

where $I(x,y)$ is the intensity of a pixel $(x,y)$, $I_0$ the background intensity, $x_g$ and $y_g$ the position of the peak, $\theta$ the rotation angle, $\sigma_x$ and $\sigma_y$ the standard deviation along two orthogonal directions. The fitting was performed automatically using CalAtom.

### Density functional theory calculations

First-principles calculations were performed using projected augmented wave (PAW) potentials with Perdew-Burke-Ernzerhof (PBE) functional generalized gradient approximation (GGA)[53] as implemented in the Vienna ab initio simulation package (VASP)[54]. A planewave-basis cutoff energy of 300 eV and the Brillouin zone was sampled by the Monkhorst-Pack grid[55]. The total energy minimization

was performed with a tolerance of 10$^{-5}$ eV and all atoms were fully relaxed until the force on each atom was less than 0.02 eV/Å.

Finer k-point meshes were used for density of state (DOS) calculations with Bloch corrected tetrahedron method. All Monkhorst-Pack scheme of the Brillouin zone with the k meshes were summarized in Supplementary Table 3. The diffusion energy barrier for Bi and Ge atoms along Te$_6$ octahedrons were calculated using the climbing image nudged elastic band method (CI-NEB)[56] implemented in VASP. The diffusion path was first constructed by linear interpolation of atomic coordinates and then relaxed until the forces on each atom were less than 0.03 eV/Å.

Bi$_2$Te$_3$ and GBT (0001) surfaces were modeled using 4 QLs and 4 SLs supercells, respectively, while the other surfaces were modeled using 10 atomic layers. 20 Å vacuum spacing for all top surface slabs were used to avoid impact of periodic structure, and 2 atomic layers at bottom surface were fixed. Surface reconstruction was not considered. The surface energy (γ) was calculated by[48,49]:

$$\gamma = \frac{1}{2A}\left[ E_{slab}^{unrelax} - nE_{bulk} \right] + \frac{1}{A}\left[ E_{slab}^{relax} - E_{slab}^{unrelax} \right]$$

where $E_{slab}^{unrelax}$ and $E_{slab}^{relax}$ are the energies of unrelaxed and relaxed surfaces, respectively, A is the surface area, $E_{bulk}$ is the bulk energy of Bi$_2$Te$_3$ and GBT unit cell, and $n$ is the unit cell number contained in the slabs.

Formation energy ($E_f$) of a surface structure with composition Ge$_x$Te$_y$ can be calculated using the following equation[16,57]:

$$E_f = \frac{E\left(Ge_xTe_y\right) - x\mu_{Ge} - y\mu_{Te}}{V}$$

where $E(Ge_xTe_y)$ is the total energy of Ge$_x$Te$_y$ layer on the surface, $\mu_{Ge}$ and $\mu_{Te}$ are chemical potential of Ge and Te, respectively. $V$ is the volume of a Ge$_x$Te$_y$ layer. The two chemical potentials are related by $\mu_{Ge} + \mu_{Te} = \varepsilon_{GeTe}$, where $\varepsilon_{GeTe}$ is the energy of a unit cell in GeTe bulk structure.

### Reporting summary

Further information on research design is available in the Nature Portfolio Reporting Summary linked to this article.

## Data availability

The data that support Figs. 2, 3 and 5 can be found in the Source Data, the data that support the other findings of this study are available from the corresponding authors upon request. Source data are provided with this paper.

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

## Acknowledgements

This work was supported by the National Natural Science Foundation of China (11834012 WY.Z.; 91963207 X.S.; 51902237 X.S.; 52272235 X.S.; 52130203 WY.Z.), National Key Research and Development Program of China (2019YFA0704903 X.S.), and Foshan Xianhu Laboratory of the Advanced Energy Science and Technology Guangdong Laboratory (XHT2020-004 WY.Z.). FESEM and EPMA experiments were performed at the Center for Materials Research and Testing of Wuhan University of Technology (WUT). The S/TEM work was performed at the Nanostructure Research Center (NRC), which is supported by the Fundamental Research Funds for the Central Universities (WUT: 2021III016GX).

## Author contributions

WY.Z., Q.Z., and X.S. conceived the idea. W.C. and X.S. designed and performed the in situ STEM experiments and analyzed the data. W.C., WC.L., WX.L. and C.L. synthesized the $GeBi_2Te_4$ samples and prepared TEM samples. W.C., Z.G., H.M. and W.Z. performed DFT simulation. W.C., G.V.T. and X.S. drafted the manuscript. All authors contributed to data analysis, result interpretation and writing the paper.

## Competing interests

The authors declare no competing interests.
