## [Peer Review File · Nature Communications]

Direct observation of cation diffusion driven surface reconstruction at van der Waals gapsReviewer #1 (Remarks to the Author):

Using the in-situ aberration-corrected ADF-STEM imaging technique, the authors gave a deep atomistic insight into the cation diffusion inside the vdW gaps and the etching of vdW surfaces at high temperatures with the intuitive Z-contrast. They claimed to observe the in-plane cation diffusion inside the vdW gaps with the atomic resolution for the first time, in addition to the complex surface reconstruction assisted by such kinds of cation diffusion. Quantitative analysis of ADF-STEM images including strain mapping and intensity/angle fitting based on the Gaussian model was conducted to unravel the details of cation diffusion and the behavior of surface etching under beam eradication and heating. This work is novel and important for the design of novel vdW materials, and it should provide valuable references for the design of novel functional vdW materials if the following problems can be further clarified:

1. In Figure 1 (d) and Supplementary Figure S5, the ADF-STEM results of the stacking fault at 100 °C are given, but the video showing the graduate appearance of the stacking faults is missing. In addition, the ADF-STEM images show a higher intensity around the stacking faults, such a contrast change around stacking faults is caused by the segregation of cations or by the channeling effect? Moreover, the central areas in Supplementary Figure S5 (b) have no appearance of vdW gaps? Is it because the Bi/Ge cations go to the vdW gaps and form a new phase?
2. In Supplementary Figure S5, the DLs are frequently observed to be shared by SL/QL layers, and the Bi/Ge atom layer must be interchanged with the Te layer in the transitional region, and this phenomenon is also observed in the in-situ result shown in Figure 2-4. Please explain the frequent appearance of such phenomenon, which should be should be key in this paper.
3. A nice quantitative analysis of the in-situ ADF-STEM data of cation diffusion is provided in Figures 2-3, and the correlation between the cation density and the local Te₆ octahedron distortion for all of the STEM snapshots. However, whether this cation diffusion is toward the right along the vdW gap or knocked off by the electron beam is not confirmed considering the 300keV energy of the incident electron and 50pA beam current. This question should be answered if using the in-situ results to instruct the design of novel vdW materials.
4. How the shearing strain in Figures 2-3 is released in the 155-165s?
5. The composition and the structure of surface DLs in the {0001}, {01-1-7}, {01-14} surfaces seem not very resolved. The current evidence such as ADF-STEM simulations is not well agreed with the experimental contrast (Figure 1 (i), {0001} surface), other evidence such as ABF-STEM, or the EDX/EELS result should be provided to provide a deeper insight into the structure and composition of DL.
6. The current ADF-STEM images in the {0117}, {0114} surfaces indicate the great flexibility of the DL structure on the surfaces. Consequently, is it possible that the DLs in Seg1 and Seg2 are the same DL? In this case, both the Te atoms and the Ge/Bi atoms are diffused around the dislocation core.
7. The authors claim that the GBT structure adjacent to the GeTe surface DL always has the QL structure, and conclude that the formation of a GeTe surface DL can stabilize the GBT QL, and vice versa. Is it frequently observed during experiments or only a specific result?

Additionally, these technical questions should be answered by the authors:

1. As widely acknowledged, the measured strain by the GPA is sensitive to the selected Bragg peaks, please clarify which pair of Bragg peaks are selected in all of the GPA analyses, and does the selection of the Bragg peaks matter in the computation of strain distributions?
2. In the GPA analysis result, the horizontal fridges appear in the E_{yy} and E_{xy} components due to the scanning distortion. Please clarify whether the shearing strain claimed in Figure 3 and Supplementary Figure 7 is the scan distortion or the intrinsic strain of the sample?
3. How the experimental ADF-STEM images shown in this paper are denoised and

filtered?

4. In the quantitative analysis of the ADF-STEM images, all of the atom columns are fitted simultaneously or the atom columns are fitted individually? If fitted simultaneously, how to deal with the background?
5. In the ADF-STEM simulation, how is the thermal diffuse scattering considered? Via the quantum excitation method or the frozen phonon? How the incoherence of the electron beam is considered?
6. In the in-situ STEM experiment, why the collection angle is set to 48-200 mrad? I think it should not be called the HAADF-STEM, but use the ADF-STEM instead.
7. In the computation of surface formation energies, is the DL considered for all types of surfaces?
8. The nomination of DLs is a little bit confusing in this paper, the surface DLs have three layers of atoms, while the DL inside the sample has two layers.

Some other minor problems including grammar mistakes and typos are marked in the attached files.

Reviewer #2 (Remarks to the Author):

In this manuscript entitled "Direct observation of cation diffusion driven surface reconstruction at van der Waals gaps", using in situ heating STEM, the authors directly observed the Ge/Bi cation diffusion along the vdW gap in layered GeBi₂Te₄ (GBT). They claimed that the cation concentration variation during diffusion was correlated with the local Te₆ octahedron distortion based on a quantitative analysis of the atomic column intensity and position in time-elapsd STEM images. The in-plane cation diffusion leads to out-of-plane surface etching through complex structural evolutions involving the formation and propagation of a non-centrosymmetric GeTe double layer (DL) surface reconstruction on fresh vdW surfaces, and GBT subsurface reconstruction from a septuple layer to a quintuple layer structure. The results regarding cation diffusion in the vdW gap are quite interesting and timely, meanwhile the data is solid and the manuscript is also well written. I will recommend it for publication in Nature Communications after the following points are addressed:

1. GeBiTe is an emerging family of topological materials similar as MnBiTe. Since the main observation is about the structural evolution under excitation, I suggest the author put more focus on how the structure could affect the topological properties of this material in the introduction or discussion part, so that the manuscript could fit in the current interest and may also improve its potential impact.
2. The dislocation core and the extra Ge/Bi sublayer shift to the left by eight Te₆ octahedrons (3.03 nm), indicating that some Ge/Bi cations have diffused away. What is the energy barrier for cation diffusion out of the Te₆ octahedrons? What is the corresponding mechanism?
3. The authors summarized the relation between the local geometric parameters and the local cation intensity of the 25 Te₆ octahedrons from the two STEM frames under strain and the three STEM frames after the strain is relaxed. In the manuscript, a linear fitting was used. Maybe more theoretical calculations are needed to support such a relationship.
4. The author claims the out-of-plane etching mainly originates from a peeling of the Seg 1 top Te sublayer to avoid the formation of Seg 1 stacking on Seg 2. I was curious why the Seg 1 cannot exist on Seg 2?

Reviewer #3 (Remarks to the Author):

Cui et al has presented the study of cation diffusion in GeBi₂Te₄ via in-situ heating STEM. They have shown atomical resolution imaging of the cation diffusion along the

vdW gap consistent with simple diffusion model. They also show imaging of DL surface reconstruction mechanism driven by cation diffusion. The paper has provided insights in understanding cation diffusion and vdW gap structural evolution in GBT materials. However, I think the authors should elaborate more on the importance and impact of this study to attract more interests from the field of vdW materials and heterostructures. Here are several questions:

- 1. Are there more observations of the cation diffusion incident to further validate the diffusion mechanism? The non-consistent diffusion lengths are explained by strain/vdW gap spacing. It requires data from more diffusion incidents to make this conclusion more convincing.**
- 2. Can the authors elaborate more on the importance of the DL surface reconstruction? How can it theoretically impact thermoelectric/electronic properties of the GBT materials?**
- 3. Is this imaging approach potentially viable to observe ion diffusion in other vdW structures, such as Li⁺ in TMDCs?**

Responses to Comments of Nature Communications 22-21516-T

We would like to thank the reviewers for the professional and valuable comments on our manuscript. The manuscript has been revised based on the reviewers' comments.

REVIEWER COMMENTS

Reviewer #1 (Remarks to the Author):

Using the in-situ aberration-corrected ADF-STEM imaging technique, the authors gave a deep atomistic insight into the cation diffusion inside the vdW gaps and the etching of vdW surfaces at high temperatures with the intuitive Z-contrast. They claimed to observe the in-plane cation diffusion inside the vdW gaps with the atomic resolution for the first time, in addition to the complex surface reconstruction assisted by such kinds of cation diffusion. Quantitative analysis of ADF-STEM images including strain mapping and intensity/angle fitting based on the Gaussian model was conducted to unravel the details of cation diffusion and the behavior of surface etching under beam eradication and heating. This work is novel and important for the design of novel vdW materials, and it should provide valuable references for the design of novel functional vdW materials if the following problems can be further clarified:

Reply: We appreciate the reviewer for the generally positive comments on our work. The reviewer's comments are addressed in the following section.

1. In Figure 1 (d) and Supplementary Figure S5, the ADF-STEM results of the stacking fault at 100 °C are given, but the video showing the graduate appearance of the stacking faults is missing. In addition, the ADF-STEM images show a higher intensity around the stacking faults, such a contrast change around stacking faults is caused by the segregation of cations or by the channeling effect? Moreover, the central areas in Supplementary Figure S5 (b) have no appearance of vdW gaps? Is it because the Bi/Ge cations go to the vdW gaps and form a new phase?

Reply: We agree with the reviewer that more details need to be provided. We have now included an *in situ* TEM movie showing the appearance of stacking faults during

heating from room temperature to 250 °C (Supplementary Movie 1). We have also added a supplementary figure (Supplementary Fig. 5) including key frames from the movie. The figure is also shown below (Fig. R1). The gradual contrast change around the stacking faults regions during heating can be observed. As STEM images can better demonstrate the contrast change, we also included comparison between STEM images acquired at room temperature, 100 °C and 250 °C in Supplementary Fig. 5 and Fig. R1.

Figure R1 Low magnification *in situ* TEM (a-c) STEM (d-f) images during heating from room temperature (RT) to 250 °C. The appearance of stacking faults are indicated by green arrows.

The higher contrast around the stacking faults is mostly likely from diffraction contrast. As the reviewer mentioned in another question, the acceptance angle of our STEM image is 48 mrad ~ 200 mrad, and therefore the inner acceptance angle is not sufficiently high to avoid diffraction contrast in STEM images. We speculate that the change of sample orientation due to bending should be the main reason for the diffraction contrast. The bending is evident from the bending contours at low-magnification (Fig. R1a-c), and the atomic resolution STEM image showing region that is sharp (perfectly on zone) and region that is blurred (slightly off-zone) within one frame (Fig. R2e). As mentioned by the reviewer, it is also possible that the contrast could come from channeling as the vdW gap closes at the stacking fault regions, which changes the spacing between atomic columns and influence the electron channeling.

To investigate segregation of cations, EDS mapping was acquired at the stacking

fault region (Fig. R2a-d). The signal from Ge, Bi and Te is uniform across the region, proving that there is no noticeable segregation of cations or anions at the stacking fault. As the formation of stacking fault occurs at relatively low temperature (100 °C), it is to be expected that the diffusion is relatively slow and diffusion length is short. However, there could be some intermixing between cations and anions between adjacent sublayers as suggested by the reviewer in the next comment.

Figure R2. EDS element maps of Ge, Bi and Te acquired from the stacking fault region. (a) The ADF-STEM image shows the bright contrast corresponding to stacking faults. (b-d) EDS maps acquired from the region in (a). (e) Atomic resolution STEM image acquired around the stacking faults.

The vdW gap indeed closes at the central area, as can be observed in the figure below (Fig. R3). We can see that the left vdW gap and right vdW gap are misaligned by one atomic layer at the stacking fault core region, where the vdW gap seems to be filled by extra atoms. Careful analysis however shows that this is visual effect, and there is no additional sublayer from cations going into vdW gaps. For example, the distance between the two Te sublayers at the vdW gap gradually decreases from left to right (green rods and arrows in Fig. R3d), and then eventually form a DL structure with a vdW gap above it. We will discuss more in the next comment on the DL structure. The relevant figures and discussion have been included in the revised manuscript (*Line 111-122*).

Figure R3. ADF-STEM images acquired from the layered defects area. Low magnification ADF-STEM image at RT (a) and 100 °C (b). (c, d) The corresponding atomic-resolved ADF-STEM images from the boxes in a and b.

2. In Supplementary Figure S5, the DLs are frequently observed to be shared by SL/QL layers, and the Bi/Ge atom layer must be interchanged with the Te layer in the transitional region, and this phenomenon is also observed in the in-situ result shown in Figure 2-4. Please explain the frequent appearance of such phenomenon, which should be key in this paper.

Reply: We agree with the reviewer that the formation of DL is a key feature in our *in-situ* experiment. The atomistic dynamics is heavily influenced by DL structure in several aspects. First, GBT crystal structure has seven sublayers, the formation of DL has cascade effect that generate QL and subsequently TL structures as shown in Fig. 4. These structures then further form stacking faults or misalignment between them, providing abundant misaligned vdW gaps that make cation diffusion possible.

Second, as suggested by the reviewer, the cations and anions in DL keeps interchanging to obtain the desired equilibrium layered structure. During the surface reconstruction in Fig. 4, as Seg 1 is gradually replaced by Seg 2, the subsurface QL should have been reduced to a TL. However, the surface reconstruction always tends to stabilize a subsurface QL structure. To avoid formation of TL, a DL is detached from the SL below and gets attached to the TL to form a new QL. The attachment and detachment of DL is accompanied by intermixing of Bi/Ge sublayer and Te sublayer.

We can understand the intermixing by plotting the I_{top}/I_{bottom} intensity ratio for every atom pair in the DL structure from left to right for different frames from 0 s to 155 s (Fig. R4). For completely disordered structure, the ratio should be close to 1. If the top layer is rich in Bi/Ge, the ratio is larger than 1; if the top layer is rich in Te, then the ratio is less than 1. For each frame, the ratio increases from left to right and crosses the ratio = 1 line. As time evolves, the crossover position moves to the left as indicated by the red arrow, suggesting that cations gradually diffuse to the top layer. The vdW gap above the top layer gradually closes, forming the QL structure as desired. We have included the discussion in the revised manuscript (Line 283-292) and included the figure as Supplementary Fig. 15.

Figure R4. The intensity ratio of atomic column pairs in DL structure from Fig. 4, for frames acquired at different time stamp. (a, b) Atomic resolution ADF-STEM images acquired at 0 s, 60 s, 105 s and 155 s. The specific DL for intensity ratio analysis is marked by a green dashed box in each image. The white arrows indicated top and bottom sublayers. The atomic column pair (marked by green rods) has one atomic column from the top sublayer and another atomic column from the bottom sublayer. (b) The I_{top}/I_{bottom} intensity ratio for every atom pair in the DL structure from left to right for different frames from 0 s to 155 s.

3. A nice quantitative analysis of the in-situ ADF-STEM data of cation diffusion is provided in Figures 2-3, and the correlation between the cation density and the local Te_6 octahedron distortion for all of the STEM snapshots. However, whether this cation

diffusion is toward the right along the vdW gap or knocked off by the electron beam is not confirmed considering the 300 keV energy of the incident electron and 50 pA beam current. This question should be answered if using the in-situ results to instruct the design of novel vdW materials.

Reply: We agree with the reviewer that it is important to discuss the beam effect on the diffusion process. We first calculated the maximum energy (E_m) transferred from the 300 kV electron beam to the sample using the following equation (*Micron* 2004, 35: 399-409; *Nature Mater.* 2021, 20(7): 951-955):

$$E_m = \frac{2ME(E+2mc^2)}{(M+m)^2c^2+2ME} \quad (1)$$

Here, M is the mass of the target atom, E the incident electron energy (300 keV in our experiment), m the rest electron mass, and c is the speed of light. We then obtain the maximum energy E_m transferred by elastic collision for Ge atoms (11.7 eV), Bi atoms (4.07 eV), and Te atoms (6.67 eV). Displacement threshold energy has been measured for Bi atom (13 eV), Ge atom (14.5 eV), and Te atom (12 eV) in their pure element crystals (*Journal of Physics F: Metal Physics* 1987, 17(12): 2365-2372; *Revue Phys. Appl.* 1980, 15(1): 15-19; *Journal of Applied Physics* 1959, 30 (8): 1235-1238). These energies are much larger than the diffusion energy barrier of Ge/Bi cations along the vdW gaps (0.53 eV for Ge and 0.81 eV for Bi at QL vdW gap, see the second comment from Reviewer 2 for more details). Therefore, although the electron beam can transfer energy comparable to the displacement threshold energy, it is much easier to activate the diffusion process, than knocking out atoms from the bulk material. Moreover, if those cations were randomly knocked out by the beam, it would be hard to obtain the intensity profile that matches Fick's law. We are therefore confident that the observed phenomenon is a diffusion process.

We have performed additional experiments to prove that the diffusion process is mainly activated by the thermal energy. First, a region containing reconstructed surface was irradiated by the electron beam at 150 °C (Fig. R5). After 4 minutes, the layered structure, vdW gaps, and the surface structure remain the same. This confirms that temperature plays more important role than the beam effect.

Figure R5. *In situ* ADF-STEM frames showing that the reconstructed surface was stable under e-beam irradiation when the temperature was kept at 150 °C. This proves that the reconstruction is activated above certain temperature.

Then, we tested if heating alone can cause such surface reconstruction. During *in situ* experiments, we sometimes quickly moved to sample areas that had not been beam irradiated, and observed similar pore structure and surface reconstruction that have already formed (Fig. R6). Therefore, the surface reconstruction and diffusion can occur without beam irradiation. The electron beam, however, can accelerate the process by providing more energy. In the revised manuscript, Fig. R5 and Fig. R6 are included as Supplementary Fig. 17 and Supplementary Fig. 18, and relevant discussion has been included in the manuscript (Line 296-300).

Figure R6. Formation of GeTe₂ TL reconstructed surfaces without electron beam irradiation. These STEM images were acquired by quickly moving to regions that have not been beam irradiated.

4. *How the shearing strain in Figures 2-3 is released in the 155-165s?*

Reply: We would like to thank the reviewer for this useful comment. As shown in Fig. 3a, the shear strain ϵ_{xy} at the vdW gap was suddenly released from 150 s to 155 s. In the revised manuscript, we added white arrows to indicate the location of the streak in Fig. 3a. The strain was released because the Ge/Bi atoms continuously diffuse out from the Te₆ octahedrons. Initially, Ge/Bi concentration was high, and the Te₆ octahedrons have height and angle that are close to Ge/Bi-occupied Te₆ octahedrons ($\alpha = 70^\circ$, $h = 340$ pm), while as Ge/Bi cations diffuse out, the geometry of the Te₆ octahedrons is close to the empty Te₆ octahedrons at the vdW gap ($\alpha = 50^\circ$, $h = 240$ pm). The relaxation causes the up-right half of the plane to move downward a little bit.

Moreover, to address this comment and Comment 3 from reviewer 2, we have performed density functional theory (DFT) calculation to optimize the geometry of Te_6 octahedrons for different Ge/Bi concentration. The results are summarized below (Fig. R7) and the details were displayed in Fig. R14, showing how the geometric parameters change as the cation concentration changes. There is an abrupt change when the cation concentration drops down to 50%, which agrees very well the sudden release of shear strain observed in Fig. 2. We have included the DFT results in Supplementary Fig. 14 and the discussion the revised manuscript (Line 244-251).

Figure R7. The DFT relation of angle/spacing and concentration in Te_6 octahedrons. (b) The relation fitting results of Te_6 octahedrons with cationic different concentration.

5. The composition and the structure of surface DLs in the $\{0001\}$, $\{01\bar{1}7\}$, $\{01\bar{1}4\}$ surfaces seem not very resolved. The current evidence such as ADF-STEM simulations is not well agreed with the experimental contrast (Figure 1 (i), $\{0001\}$ surface), other evidence such as ABF-STEM, or the EDX/EELS result should be provided to provide a deeper insight into the structure and composition of DL.

Reply: We agree with the reviewer that it would be better to have more evidence to support the structure and composition of DL in the $\{0001\}$, $\{01\bar{1}7\}$, $\{01\bar{1}4\}$ surfaces. First, we have performed EDS mapping at a DL-reconstructed surface (Fig. R8). A yellow dashed line was drawn at the same location on Ge, Bi, Te EDS maps, and we can see that Ge and Te signal is much stronger than Bi signal outside of that yellow dashed line. EDS line profile across the interface shows that the edge is significantly rich in Te, confirming that the DL structure mainly consists of Te_6 octahedrons, which agree well the STEM intensity analysis in Fig. 1f. The EDS line profile also suggests that Ge signal is stronger than Bi at the edge. Combining this with the very weak contrast from the cations in DL, we can deduce that the DL indeed has the proposed structure. Note that as we stated in the manuscript, we cannot rule out that there are

some residue Bi atoms in the surface DL.

Second, we have performed structural optimization using density functional theory to explore the stability of the DL structure. We constructed a GeTe DL on Bi₂Te₃ QL and the optimized structure agrees quite well with the observed STEM image in Fig. 1f-i.

The EDS results and the DFT simulation results have been added as Supplementary Fig. 9. The comparison between experimental STEM image and the optimized surface structure is included in Fig.1f of the revised manuscript. Relevant discussion has also been included in the revised manuscript (Line 137 and 156).

Figure R8. The composition and crystal structure of reconstructed GeTe₂ TL surface revealed by EDS and density functional theory (DFT). (a) EDS maps of Ge, Bi and Te element at a reconstructed (0117) surface, with the yellow dashed line marked the interface between GBT and the GeTe₂ TL surface. (b) The line profile of EDS intensity along the white line in ADF-STEM image (a). (c) The interface structure between GeTe₂ TL and GBT QL was relaxed using DFT. The relaxed structure matches well with the experimental STEM image as shown in Fig. 1f.

6. The current ADF-STEM images in the {0117}, {0114} surfaces indicate the great flexibility of the DL structure on the surfaces. Consequently, is it possible that the DLs in Seg1 and Seg2 are the same DL? In this case, both the Te atoms and the Ge/Bi atoms are diffused around the dislocation core.

Reply: We would like to thank the reviewer for this interesting comment. The DL structure is indeed flexible as it seamlessly attaches on the surface regardless of the steps and other surface defect as can be seen from Fig. 4 and Supplementary Fig. 23. Based on the STEM images, it is hard to tell whether the outermost Te sublayer diffuses

along the surface, or the inner Te sublayer diffuses around the dislocation core as suggested by the review. There are two possible reasons that we prefer the first case. First, in some frames, we can see Seg1 seems to collapse as indicated by green arrows in Fig. R9 below. That means Seg1 seems to become unstable as Seg2 forms underneath. Second, we performed DFT calculation on possible diffusion of Te anions along the vdW gap. The Te anion can cause the vdW gap to increase from 260 pm to 340 pm for Bi₂Te₃, and from 260 pm to 380 pm for GBT (Fig. R9d, e). We should be able to capture such big change of vdW gap size in STEM images if Te anions diffuse along the vdW, but it has never been observed in our experiments. We therefore believe it is more likely for Seg 1 to collapse from the surface.

Figure R9. Collapse of Seg 1 induced by cations diffusion along vdW gap. (a-c) ADF-STEM images of the evolution of seg 1 and Seg 2 structure acquired at 145 s, 150 s and 155 s at 400 °C. (d, e) The DFT optimized Bi₂Te₃ and GBT 3×2×1 supercell structure with Te anion in the vdW gap.

7. The authors claim that the GBT structure adjacent to the GeTe surface DL always has the QL structure, and conclude that the formation of a GeTe surface DL can stabilize the GBT QL, and vice versa. Is it frequently observed during experiments or only a specific result?

Reply: Yes, whenever there is a DL reconstruction on (0001) surface, the GBT SL structure always changes to QL structure. We have added more examples as can be seen in Fig. R17-19. It has been reported that the surface reconstruction, regardless of the formation of DL, always causes formation of QL structure. For example, the paper by Hou et al. states that the MnBi₂Te₄ surface structure is terminated by amorphous layer

and a QL structure (Fig. R10a, from Fig. 1g in *ACS Nano* 2020, 14(9): 11262-11272). The formation of subsurface QL was also observed in GeBi_2Te_4 (Fig. R10b, from Fig. 4-11 in *Callaert C. Characterization of defects, modulations and surface layers in topological insulators and structurally related compounds[D]. University of Antwerp, 2020*). The additional examples from our experiments are now included as Supplementary Figures 20-22.

Figure R10. Other examples showing surface reconstruction on QL structure. (a) $(000l)$ surface terminated with MnTe amorphous layer on a QL in MnBi_2Te_4 . (b) $(000l)$ surface reconstruction in GeBi_2Te_4 due to slightly oxidation.

Additionally, these technical questions should be answered by the authors:

1. As widely acknowledged, the measured strain by the GPA is sensitive to the selected Bragg peaks, please clarify which pair of Bragg peaks are selected in all of the GPA analyses, and does the selection of the Bragg peaks matter in the computation of strain distributions?

Reply: We agree with the reviewer that more details need to be provided for GPA. Here, the selected Bragg peaks were $(000l)$ -type diffraction spot and $(01\bar{1}\bar{7})$ diffraction spot. In Fig. R11 below, we have drawn $(000l)$ and $(01\bar{1}\bar{7})$ crystallographic planes on the STEM image, and we can see atoms are densely packed in those planes. Therefore, the best results were obtained using one $(000l)$ -type diffraction spot and $(01\bar{1}\bar{7})$ diffraction spot for Fig. 3 and Supplementary Fig. 8 and Supplementary Fig. 12.

We then used different combinations of diffraction spots such as (0006) , $(01\bar{1}\bar{7})$, $(01\bar{1}5)$, $(01\bar{1}1)$ and $(01\bar{1}\bar{1}\bar{1})$ for GPA strain maps (Fig. R11d-i). The general conclusion is that, to obtain consistent results, a $(000l)$ -type diffraction spot needs to be selected. The details on GPA diffraction spots choice are now included in the ‘Methods’ section.

Figure R11. GPA strain maps of the same region using different pairs of diffraction spots. (a) Simulated electron diffraction pattern along $[\bar{2}110]$ zone axis. The diffraction spots used for testing are marked by circles with different colors. (b) An ADF-STEM image acquired from the region with layered defects. (c) Atomic resolution ADF-STEM image and the crystallographic planes corresponding to different diffraction spots in (a). (d-i) GPA strain maps using different combinations of diffraction spots as labeled above each image.

2. In the GPA analysis result, the horizontal fridges appear in the E_{yy} and E_{xy} components due to the scanning distortion. Please clarify whether the shearing strain claimed in Figure 3 and Supplementary Figure 7 is the scan distortion or the intrinsic

strain of the sample?

Reply: We agree with the reviewer that the similarity between the shear strain and the scan distortion needs to be clarified. STEM images generally suffer from high frequency distortion such as jittering and noise, and low-frequency drift distortion. The distortion from jittering causes misalignment between two adjacent rows, which can cause random local contraction, expansion or shearing. Due to the randomness nature of jittering, it is very unlikely that jittering can cause systematic evolution of the shear strain around the dislocation line as shown in Fig. 3. Moreover, jittering tends to be constant for one entire horizontal scan line. The shear strain caused by jittering should range from the leftmost to the rightmost of the frame, while here the shear strain only starts from the dislocation core. Low-frequency drift distortion can be approximated by 2D linear distortion for small dwell time. 2D linear distortion can introduce global strain, but not local shear strain as shown in Fig. 3. Therefore, the observed shear strain should be intrinsic. We have added relevant discussion in the revised manuscript (*Line 225-228*).

3. How the experimental ADF-STEM images shown in this paper are denoised and filtered?

Reply: All STEM images shown in Fig. 1-4 were first background-corrected by filtering out low-frequency information around the (000) diffraction spot, and then blurred using a 2D gaussian distribution with $\sigma = 1$, implemented in the commercial software Velox that comes with Titan Themis microscope. For quantitative analysis, the raw data of STEM images were exported from Velox, denoised using a gaussian filter with $\sigma = 2$, and then analyzed using the free package CalAtom_v1.2. The frames in the supplementary movies were batch-filtered using a Gaussian distribution with $\sigma = 1$. We have included this information in the “Methods” section.

4. In the quantitative analysis of the ADF-STEM images, all of the atom columns are fitted simultaneously or the atom columns are fitted individually? If fitted simultaneously, how to deal with the background?

Reply: The atomic columns were fitted individually using Gaussian distribution. For each atomic column, the pixels containing the atomic column were fitted using a

Gaussian distribution as defined below:

$$I(x, y) = I_0 + A \exp \left\{ - \frac{[(x - x_g) \cos \theta - (y - y_g) \sin \theta]^2}{2\sigma_x^2} - \frac{[(x - x_g) \sin \theta + (y - y_g) \cos \theta]^2}{2\sigma_y^2} \right\}$$

where $I(x, y)$ is the intensity of a pixel (x, y) , I_0 the background intensity, x_g and y_g the position of the peak, θ the rotation angle, σ_x and σ_y the standard deviation along two orthogonal directions. The fitting was performed automatically using a free software called CalAtom (*Ultramicroscopy* 2019, 202: 114-120). For example, in the figure below, an atomic column is indicated by a white arrow, and the pixels that are used for Gaussian fit are encircled by a green circle. As the fitting was performed on individual atomic column, the background was automatically taken care of. We have included this information in the “Methods” section.

Figure R12. The green circle indicates the pixels used for Gaussian distribution fitting, for a single atomic column.

5. In the ADF-STEM simulation, how is the thermal diffuse scattering considered? Via the quantum excitation method or the frozen phonon? How the incoherence of the electron beam is considered?

Reply: In the manuscript, the STEM images were simulated using multi-slice algorithm implemented in QSTEM package. The TDS was considered using the frozen phonon method with 10 configurations. The incoherence of the electron beam was considered by applying extra blurring to the simulated images until the best match between simulation and experiment is achieved. We have included the details in the Methods section of the revised manuscript.

6. In the in-situ STEM experiment, why the collection angle is set to 48-200 mrad? I think it should not be called the HAADF-STEM, but use the ADF-STEM instead.

Reply: We would like to thank the reviewer for mentioning this. We agree with the reviewer that conventionally the inner acceptance angle should be higher than 50 mrad to be considered as HAADF-STEM (see for example, David B. Williams & C. Barry Carter. *Transmission Electron Microscopy-A Textbook for Materials Science*. Springer, 1996: 379-380). The collection angle on our microscope is 48-200 mrad when the camera length is 115 mm, which is recommended since the training. We did not pay much attention until the paper was written. The results should be fine as long as the QSTEM simulation uses the correct experimental parameters. We have changed the term from HAADF-STEM to ADF-STEM in the revised manuscript.

7. In the computation of surface formation energies, is the DL considered for all types of surfaces?

Reply: We did not consider the DL reconstruction to calculate surface formation energies. The inclusion of DL in the computation would be difficult for $\{01\bar{1}\bar{7}\}$ and $\{01\bar{1}4\}$ surfaces due to the steps on the surface and the flexibility of the DL. We speculate that the general trend should not be reverted by the DL reconstruction. We have clarified this point in the revised manuscript (*Line 442*).

8. The nomination of DLs is a little bit confusing in this paper, the surface DLs have three layers of atoms, while the DL inside the sample has two layers.

Reply: We agree with the reviewer that the nomination DL is confusing as the structure indeed has three layers of atoms, which should be defined as triple layer (TL). The term GeTe is not accurate either, as it indeed contains two Te sublayers and one Ge sublayer. To avoid confusion, we have changed the nomination from 'GeTe DL' to 'GeTe₂ TL' in the revised manuscript. In the response letter, we decide to use 'GeTe DL' for consistency, so that the reviewers are not confused.

Some other minor problems including grammar mistakes and typos are marked in the attached files.

Reply: We would like to thank the reviewer for taking your time to correct the grammar mistakes and typos. They have been corrected as suggested by the reviewer in the revised manuscript.

Reviewer #2 (Remarks to the Author):

In this manuscript entitled “Direct observation of cation diffusion driven surface reconstruction at van der Waals gaps”, using in situ heating STEM, the authors directly observed the Ge/Bi cation diffusion along the vdW gap in layered GeBi₂Te₄ (GBT). They claimed that the cation concentration variation during diffusion was correlated with the local Te₆ octahedron distortion based on a quantitative analysis of the atomic column intensity and position in time-elapsd STEM images. The in-plane cation diffusion leads to out-of-plane surface etching through complex structural evolutions involving the formation and propagation of a non-centrosymmetric GeTe double layer (DL) surface reconstruction on fresh vdW surfaces, and GBT subsurface reconstruction from a septuple layer to a quintuple layer structure. The results regarding cation diffusion in the vdW gap are quite interesting and timely, meanwhile the data is solid and the manuscript is also well written. I will recommend it for publication in Nature Communications after the following points are addressed:

Reply: We would like to thank the reviewer for the generally positive comments. We have performed DFT calculations to answer the reviewer’s comments as follows.

1. GeBiTe is an emerging family of topological materials similar as MnBiTe. Since the main observation is about the structural evolution under excitation, I suggest the author put more focus on how the structure could affect the topological properties of this material in the introduction or discussion part, so that the manuscript could fit in the current interest and may also improve its potential impact.

Reply: We agree with the reviewer that discussion on the topological properties could improve the potential impact of this manuscript. GBT is a topological insulator with bulk energy gap ~ 180 meV. The topological surface state below and above the Dirac point are isolated from the bulk band. It has been reported that the disorder in GBT has little effect on the surface state spin polarization (*Phys Review B* 2012, 86: 195304). However, it is to be expected that the drastic surface reconstruction shown here changes the surface chemical composition, the stacking number, and possibly the majority point defect and the majority carrier, which can have an influence on the electric and spin

properties. This can further change the relative contribution of topological surface states to the electron transport. We have added relevant discussion in the revised manuscript (*Line 299-309*).

2. *The dislocation core and the extra Ge/Bi sublayer shift to the left by eight Te₆ octahedrons (3.03 nm), indicating that some Ge/Bi cations have diffused away. What is the energy barrier for cation diffusion out of the Te₆ octahedrons? What is the corresponding mechanism?*

Reply: We agree with the reviewer that the diffusion mechanism needs to be elucidated. The diffusion energy barrier for Bi and Ge atoms along the Te₆ octahedrons were calculated using the climbing image nudged elastic band method (CI-NEB) implemented in VASP. The diffusion path was first constructed by linear interpolation of atomic coordinates and then relaxed until the forces on all atoms were less than 0.03 eV/Å (the details are now included in Methods section of the revised manuscript). The calculation results are shown in Fig. R13, and included in SI as Supplementary Fig. 10. The energy barrier of Ge (Bi) atoms diffused in vdW gap of Bi₂Te₃, GBT are 0.53 eV (0.81 eV), and 0.42 eV (0.27 eV), respectively (Table R1). At 400 °C, the thermal energy is $k_B T = 0.06$ eV. The electron beam irradiation could provide additional energy that facilitates the diffusion process. The energy barrier for Bi cation to diffuse along GeBi₂Te₄ is significantly lower than that in Bi₂Te₃, which is probably why this phenomenon is observed in the GeBi₂Te₄ system instead of the Bi₂Te₃. For Ge diffusion in GeBi₂Te₄ and Bi₂Te₃, and Bi diffusion in Bi₂Te₃, the diffusion path is very similar. The Ge/Bi cation at the center of a Te₆ octahedron diffuses into the adjacent Te₄ tetrahedron, and eventually to the adjacent Te₆ octahedron. The diffusion of Bi atom in GeBi₂Te₄ is different that the Bi atom directly diffuses between adjacent Te₆ octahedrons, resulting in smaller diffusion energy barrier (Fig. R13e, f). We have added the discussion on the diffusion barrier in the revised manuscript (*Line 188-192*).

Table R1. The diffusion energy barrier of Ge and Bi atom along the vdW gaps of Bi₂Te₃ and GBT layered structures.

Cation type	Diffusion energy barrier (eV)
	vdW gap (Bi ₂ Te ₃)
	vdW gap (GBT)

Ge	0.53	0.42
Bi	0.81	0.27

Figure R13. The diffusion energy barrier of Ge and Bi atoms in vdW gaps of Bi_2Te_3 and GeBi_2Te_4 . (a, b) The migration path of Ge and Bi atoms diffusing along the vdW gaps of Bi_2Te_3 (a) and GeBi_2Te_4 (b) layered structure respectively. The calculation was performed using the climbing image nudged elastic band method (CI-NEB) implemented in VASP. $3 \times 2 \times 1$ supercells of Bi_2Te_3 and GeBi_2Te_4 were used in the calculation. The green and red octahedrons indicate initial-state and final-state octahedrons. The energy barriers of Ge and Bi cation diffusion in vdW gap of Bi_2Te_3 and GeBi_2Te_4 are plotted in (c) and (d), respectively. (e) The diffusion path of Ge cation in Bi_2Te_3 vdW gap. The Ge cation first diffuses into the adjacent blue tetrahedron and then the red octahedron. (f) The diffusion path of Bi cation in GBT vdW gap. The Bi cation directly diffuses from the initial green Te_6 octahedron to the adjacent red Te_6 octahedron without entering the Te_4 tetrahedron.

3. The authors summarized the relation between the local geometric parameters and the local cation intensity of the 25 Te_6 octahedrons from the two STEM frames under strain and the three STEM frames after the strain is relaxed. In the manuscript, a linear fitting was used. Maybe more theoretical calculations are needed to support such a relationship.

Reply: We would like to thank the reviewer for the useful suggestion. To investigate the influence of cation concentration on the geometric parameters of Te_6 octahedrons, we create seven $2 \times 6 \times 1$ GBT supercells with different number of cations at a particular atomic column in the Ge sublayer. When projected along the [100] zone axis, that

atomic column consists of six cation sites. The initial supercell has 6 Bi atoms in the atomic column. We then gradually decrease the number of cations, creating supercells with 6 Bi cations, 5 Bi and 1 Ge cations, 4 Bi and 2 Ge cations, 3 Bi and 2 Ge cations, 2 Bi and 2 Ge cations, 2 Bi and 1 Ge cations, 1 Bi and 1 Ge cations, 1 Ge cation, and no cation occupancy at the site. The supercells were then relaxed using density functional theory. The total energy minimization was performed with a tolerance of 10^{-6} eV and fully relaxed until the force on each atom was less than 0.02 eV/Å. The α and h values as a function of number of cations are plotted in Fig. R14. We can see as the cation number decreases, both geometric parameters decrease, which agree well with our experimental results. We have added relevant discussion in the revised manuscript (Line 244-248).

Figure R14. DFT calculation showing how the average geometric parameters of the projected Te_6 octahedrons are influenced by cation concentration of the atomic column. (a) The atomic structure model of a $2 \times 6 \times 1$ GBT supercell. When projected along the b axis, the STEM image comes from the overlap of six Te_6 octahedrons. We now focus on the octahedron framed by the black square. The arrangement of this octahedron along b axis is shown next to the black square. (b) The (α, h) parameters for different combination of cations along the b axis, including 6 Bi cations, 5 Bi and 1 Ge cations, 4 Bi and 2 Ge cations, 3 Bi and 2 Ge cations, 2 Bi and 2 Ge cations, 2 Bi and 1 Ge cations, 1 Bi and 1 Ge cations, 1 Ge cation, and no cation occupancy. Both the spacing and the angle changes abruptly when the cation concentration is around 50%. (c) The crystal structure models of

GBT supercells with different cation concentration.

4. The author claims the out-of-plane etching mainly originates from a peeling of the Seg 1 top Te sublayer to avoid the formation of Seg 1 stacking on Seg 2. I was curious why the Seg 1 cannot exist on Seg 2?

Reply: We agree with the reviewer that it is important to discuss why Seg 2 and Seg 1 cannot coexist. The surface reconstruction with only Seg 1 or Seg 2 can be referred as GeTe₂ TL reconstruction, while the reconstruction with Seg 1 on Seg 2 can be referred Ge₂Te₃ QL reconstruction. Throughout *in situ* STEM experiment, we have not observed Ge₂Te₃ QL reconstruction. More instances of GeTe₂ TL reconstruction can be found in Fig. R16-18 and Supplementary Fig. 20-22. We then performed DFT simulation to understand how the Te chemical potential can influence the formation energy of GeTe₂ TL reconstruction and Ge₂Te₃ QL reconstruction change as a function of (Fig. R15). Formation energy (E_f) of a surface structure with composition Ge_xTe_y can be calculated using the following equation:

$$E_f = \frac{E(\text{Ge}_x\text{Te}_y) - x\mu_{\text{Ge}} - y\mu_{\text{Te}}}{V}$$

Where $E(\text{Ge}_x\text{Te}_y)$ is the total energy of Ge_xTe_y layer on the surface, μ_{Ge} and μ_{Te} are chemical potential of Ge and Te, respectively. V is the volume of a Ge_xTe_y layer. The two chemical potentials are related by $\mu_{\text{Ge}} + \mu_{\text{Te}} = \varepsilon_{\text{GeTe}}$, where $\varepsilon_{\text{GeTe}}$ is the energy of a unit cell in GeTe bulk structure. The equation then changes to:

$$E_f = \frac{E(\text{Ge}_x\text{Te}_y) - x\varepsilon_{\text{GeTe}} - (y - x)\mu_{\text{Te}}}{V}$$

We can see that as the Te chemical potential increases, the formation of GeTe₂ TL is preferred. As EDS mapping reveals, the edge is rich in Te. This is probably why Seg 2 cannot stack on Seg 1 to form a QL structure. we have included relevant discussion in the revised manuscript (*Line 324-328*).

Figure R15. Formation energies of the GeTe₂ TL and Ge₂Te₃ QL as a function of Te chemical potential (μ_{Te}). Here, μ_{Te} in Te₃ bulk ($\mu_{\text{Te}@\text{Te}_3_bulk} = -3.142$ eV) is set as reference energy.

Reviewer #3 (Remarks to the Author):

Cui et al has presented the study of cation diffusion in GeBi₂Te₄ via in-situ heating STEM. They have shown atomical resolution imaging of the cation diffusion along the vdW gap consistent with simple diffusion model. They also show imaging of DL surface reconstruction mechanism driven by cation diffusion. The paper has provided insights in understanding cation diffusion and vdW gap structural evolution in GBT materials. However, I think the authors should elaborate more on the importance and impact of this study to attract more interests from the field of vdW materials and heterostructures.

Reply: We would like to thank the reviewer for the very useful comments to improve the manuscript.

Here are several questions:

1. Are there more observations of the cation diffusion incident to further validate the diffusion mechanism? The non-consistent diffusion lengths are explained by strain/vdW gap spacing. It requires data from more diffusion incidents to make this conclusion more convincing.

Reply: We agree with the reviewer that more data needs to be included to understand the diffusion mechanism. We have now included three more cases showing the cation diffusion induced surface reconstruction (Fig. R16-18). Similar to Fig. 2 and Fig. 4, the dislocation cores are indicated by cyan arrows. In all three cases, we can see the movement of dislocation core, and the formation of GeTe DL on the surface. However, it is very difficult to capture the sudden change of the vdW gap distance and the shear

strain release, because normally the structure collapses before we can take an *in situ* movie. We have now included DFT calculations that reveal the diffusion path, diffusion energy barrier, and the Te₆ octahedron distortions in the revised manuscript. These three figures are now included in SI as Supplementary Fig. 20-22.

Figure R16. Additional example showing surface reconstruction process induced by cations diffusion. The cyan and white dotted rods mark the QLs and SLs rods. The cyan arrows and green lines indicate the dislocation cores and the Ge/Bi extra half planes.

Figure R17. Additional example showing surface reconstruction process induced by cations diffusion. The cyan arrows and green lines indicate the dislocation cores and the Ge/Bi extra half plane. The yellow arrow marks the step at $(01\bar{1}7)$ crystallography plane.

Figure R18. Additional example showing surface reconstruction process induced by cations diffusion. The cyan and yellow arrows and indicate the dislocation cores and the steps at (0001) surface.

2. Can the authors elaborate more on the importance of the DL surface reconstruction? How can it theoretically impact thermoelectric/electronic properties of the GBT materials?

Reply: We agree with the reviewer that the importance of DL surface reconstruction and its impact on thermoelectric properties need to be discussed. From the structure point of view, the DL surface reconstruction and cation diffusion are important as they reveal the complex structural evolution at vdW gaps with weak bonding strength, while previous investigation on the surface reconstruction of vdW layered materials has mainly focused on 2D materials edge reconstruction driven by the much stronger dangling covalent bonds and local chemistry variation. The work here can potentially help reveal structural change of other vdW layered materials during diffusion and surface reconstruction.

The DL surface reconstruction could potentially influence topological and electrical properties. We have discussed the possible influence of DL reconstruction on topological properties when addressing the second comment from Reviewer 2. As for the thermoelectric properties, GBT belongs to a large family of thermoelectric materials with the form $(\text{GeTe})_x(\text{Bi}_2\text{Te}_3)_y$, where Ge can be replaced by Sn and Pb, and Bi replaced by Sb. The GBT used in our manuscript has $x = 1$, and $y = 1$. It has been reported that GBT is p type semiconductor for $x/y > 1$ and n type semiconductor for $x/y < 1$ (*J. Alloys Compd.* 2001, 329: 50-62). Generally, as x/y ratio increases, electrical conductivity and thermal conductivity of $(\text{GeTe})_x(\text{Bi}_2\text{Te}_3)_y$ increase. The thermoelectric properties also critically depend on slight variation in stoichiometry, the main point defect type, and the microstructure. We speculate there are two possible ways the DL structure can impact the thermoelectric properties. DFT calculated density of state (DOS) of bulk GeTe and GeTe DL respectively show semiconductor to metal behavior (Fig. R19). Therefore, the DL structure on the surface has higher electrical conductivity than the bulk material, which may provide a method to increase electrical conductivity without improving thermal conductivity, which is critical for the enhancement of ZT value. The DL structure is ferroelectric, which means they can spontaneously generate local electrical field, which may introduce extra scattering to the carriers and improve Seebeck coefficient. The above discussion is now included in the revised manuscript (Line 304-316).

Figure R19. Density of state (DOS) of bulk GeTe and GeTe₂ TL. The Fermi level is set to zero.

3. Is this imaging approach potentially viable to observe ion diffusion in other vdW structures, such as Li⁺ in TMDCs?

Reply: We agree with the reviewer that it would be useful to discuss the possibility of using this method to observe Li diffusion. *In situ* STEM imaging is especially suitable for the observation of Bi atoms due to their high atomic number and strong STEM intensity. The diffusion process observed in GBT is slow enough to be captured by *in situ* STEM with temporal resolution around 1 s. Compared with Bi, there are two reasons that make observation of Li diffusion in TMDC significantly more difficult. First, the atomic number of Li⁺ (3) is much lower than typical elements in TMDC such as Mo (42), Se (34) and S (16). The weak STEM signal from Li⁺ is difficult to capture. Second, the Li⁺ ions are much smaller than Bi cation, which means the diffusion rate of Li⁺ is much faster than Bi. That means we might need higher temporal resolution to capture the diffusion process. So far, *in situ* S/TEM, EDS, EELS, and electron diffraction have been used to observe Li diffusion (*Adv. Mater.* 2019, 31(29): 1805889; *Nature Commun.* 2016, 7(1): 1-9; *Adv. Funct. Mater.* 2021, 31(16): 2010291), but direct observation at atomic scale has been very difficult.

To realize direct observation of Li diffusion, first, we should find ways to enhance Li signal in STEM images. Several TEM techniques have been developed to visualize light elements. For example, the Li signal can be captured on annular bright field (ABF) STEM images using lower acceptance angle. iDPC-STEM uses the change of the mass center of the diffraction disc to detect minute change in the projected potential, which could be useful to detect light elements (*Nature* 2021, 592(7855): 541-544.).

Second, we need ways to lower the Li diffusion rate or increase the temporal resolution of STEM imaging. Cryo-TEM can be used to lower kinetics, which might be able to make Li ions move slower (*Science* 2017, 358(6362): 506-510). The temporal

resolution of STEM imaging could be significantly improved using compressed sensing technique combined with big data and deep learning algorithms (*Microscopy and Microanalysis* (2018), 24, 623-633).

Third, it is possible use the local lattice distortion caused by Li diffusion to investigate the variation of local lithium concentration, similar to the correlation between Bi/Ge cation concentration and geometric parameters of local Te₆ octahedrons.

We have added relevant discussion in the revised manuscript (*Line 248-251*).

Reviewer #1 (Remarks to the Author):

Most of the questions I raised in the first round have been well answered except for the following questions:

1. In the STEM simulation, 10 configurations of frozen phonon are adopted, but what's the amplitude of vibration?
2. In Figure 1(c), the atom-resolved EDX maps are provided, while it seems they are horizontally flipped compared to Figure 1(b), please confirm if the flip exists and mix these three maps.
3. Frankly, the contrast of stacking fault provided in Supplementary Fig 5 and Movie 1 is hard to be seen. Please provides the data with a smaller field of view and obvious contrast.
4. The cation diffusion is always accompanied by the evolution of DL, and DL seems to have a high density even at 100 °c. I am expecting a deeper insight into the driving force for the formation and rotation of DL since studying the atomic-scale evaluation of DL should be important for the design of novel vdW materials.

Several new questions and suggestions came to mind after reading the revised manuscript, I will like to suggest the acceptance of this paper if the following problem can be answered.

1. In figure 2, please mark the TL, QL, DL, and vdW gaps on both left and right sides to help the readers to understand the structure.
2. The origin sites of pores are randomly distributed or corrected to the defects such as stacking faults in the thin film?

Reviewer #2 (Remarks to the Author):

I have gone through the whole rebuttal letter and I feel the authors have made satisfactory replies to all of my concerns and other reviewers. There are new experimental and theoretical results added, which substantially improved the manuscript. Reconstructions of topological materials are a topic that raising increasing interest. Therefore I would like to recommend its publication in Nature Communications.

I have gone through the review from reviewer #3. For my personal opinion, the authors have already presented a systematic study elaborating the experimental observation and also its theoretical interpretation. The data are consistent, the experiments are nicely executed. In the last round of review, the authors have provided additional data both experimentally and theoretically, thus I think the story is quite complete. Though this is only done in the GBT material system, it has already demonstrated a unique case of structural reconstruction induced by ion diffusion. On this basis, I think the comment #1 and #3 are a bit picky to the authors, though #2 is reasonable since the authors can elaborate more on the impact of their studies.

Responses to Comments of Nature Communications NCOMMS-22-21516A

We would like thank the reviewers for the valuable comments on our manuscript again. The manuscript has been revised based on the reviewers' comments.

REVIEWER COMMENTS

Reviewer #1 (Remarks to the Author):

Most of the questions I raised in the first round have been well answered except for the following questions:

1. In the STEM simulation, 10 configurations of frozen phonon are adopted, but what's the amplitude of vibration?

Reply: We agree with the reviewer that the amplitude of vibration needs to be provided. In the software QSTEM that we used in the manuscript, the default DW factor for an element A is calculated as $\frac{B_{Si}M_A}{M_{Si}}$, where $B_{Si} = 0.45$ is the DW factor of Si at room temperature, and M_A and M_{Si} are the atomic mass of element A and Si, respectively. We adopted this approach in the previous version of the manuscript, which gives us DW factors 0.1969, 0.0759 and 0.1212 Å² for Ge, Bi and Te, respectively. To be more precise, in the revised manuscript, we performed the frozen phonon STEM simulation using experiment DW factors ($B_{Ge}=2.21$ Å², $B_{Bi}=2.03$ Å² and $B_{Te}=1.06$ Å²) measured from single crystal X-ray diffraction (*Phys. Rev. B* 2013, 88(16): 165128). The results are not very different from previous calculation. For example, Bi concentration at Ge sublayer is estimated to be around 54 % using the new set of DW factors, compared to 53% using previous calculation. We have updated Supplementary Fig. 3 and 4 using the quantitative results based on the experimental DW factors. The description of experimental DW factors is also included in the Method section.

2. In Figure 1(c), the atom-resolved EDX maps are provided, while it seems they are

horizontally flipped compared to Figure 1(b), please confirm if the flip exists and mix these three maps.

Reply: The EDS maps are indeed flipped compared to Figure 1b. The EDS maps were actually acquired from a different STEM image. In the revised figure we have added the corresponding STEM image and the mixed EDS map in Fig. 1c. The revised part of Figure 1 is also shown below as Fig. R1.

Figure R1 Atomic resolution EDS maps of Ge, Bi and Te element in GBT SL structure.

3. Frankly, the contrast of stacking fault provided in Supplementary Fig 5 and Movie 1 is hard to be seen. Please provides the data with a smaller field of view and obvious contrast.

Reply: We agree with the reviewer that the contrast from the stacking faults could be improved by using a smaller field of view. We have updated Supplementary Fig. 5 using the reviewer's suggestion. The revised figure is also included below as Fig. R2 for your information.

Figure R2 Low magnification *in situ* TEM (a-c) STEM (d-f) images during heating from room temperature (RT) to 250 °C. Regions with stacking faults are indicated by green ovals.

4. The cation diffusion is always accompanied by the evolution of DL, and DL seems to have a high density even at 100 °C. I am expecting a deeper insight into the driving force for the formation and rotation of DL since studying the atomic-scale evaluation of DL should be important for the design of novel vdW materials.

Reply: We agree with the reviewer that it is important to understand the formation mechanism of DL structure. The formation of DL is most likely driven by the mixing between Ge/Bi-sublayer and Te-sublayer. As shown in Supplementary Fig. 15 and Fig. R3, for the left part of DL, the top sublayer is weaker than the bottom sublayer, and therefore a vdW gap appears above the top sublayer (indicated by two cyan arrows in Fig. R3b). On the other hand, for the right part of DL, the bottom sublayer is weaker than the top sublayer, and a vdW gap is below the bottom sublayer (indicated by yellow arrows in Fig. R3b). The gradual change of the intensity ratio between top and bottom sublayers is also accompanied by the propagation of vdW gap. The underlying reason that drives the intermixing between adjacent sublayers in this case might be from Ge-rich surface that changes local chemistry, but it is very difficult to probe, and is probably beyond the scope of this manuscript. We have updated Supplementary Fig. 15 to include

the discussion on the relationship between the intensity ratio and the vdW gap.

Figure R3 The intensity ratio of atomic column pairs in DL structure from Fig. 4, for frames acquired at different time stamp. (a, b) Atomic resolution ADF-STEM images acquired at 0 s, 60 s, 105 s and 155 s. (c) The I_{top}/I_{bottom} intensity ratio for every atom pair in the DL structure from left to right for different frames from 0 s to 155 s. When the intensity ratio is less than 1, the vdW gap tends to appear above the top sublayer. When the intensity ratio is larger than 1, the vdW gap tends to appear below the bottom sublayer.

We have noticed that DL can form in other QL or SL vdW layered materials. To better understand the DL formation mechanism, we are now working on DL formation in metal/ Bi_2Se_3 and metal/ Bi_2Te_3 interfaces, where we can better control the chemical potential gradient. Moreover, chemical information is easier to be extracted from binary systems using quantitative STEM, especially for Bi_2Se_3 with huge atomic number difference. In comparison, GeBiTe is not an easy system to quantify solely using STEM

intensity, because Ge and Bi cations are mixed. Ge atom columns have lower STEM intensity than Te, while Bi atom columns have higher STEM intensity than Te. Therefore, we appreciate that that the reviewer emphasizes on the importance of DL structure, and we will focus on its formation mechanism with much more details in a future manuscript.

Several new questions and suggestions came to mind after reading the revised manuscript, I will like to suggest the acceptance of this paper if the following problem can be answered.

1. In figure 2, please mark the TL, QL, DL, and vdW gaps on both left and right sides to help the readers to understand the structure.

Reply: We agree with the reviewer that labeling TL, QL, DL and vdW gaps can help the readers to better understand the structure. In the revised Fig. 2b, SLs, QLs, and DLs structures are indicated by white, cyan and green rods, respectively. VdW gaps are indicated by yellow arrows.

2. The origin sites of pores are randomly distributed or correlated to the defects such as stacking faults in the thin film?

Reply: We would like to thank the reviewer for this interesting question. From what we have observed, it seems that the pores form at the regions with smaller thickness, which could be caused by FIB sample preparation. The figure below shows that the pores are mostly formed around the center of the FIB sample, because the central region is thinner.

Figure R4 Low magnification STEM image after and before *in situ* heating.

Another example shown in the figure below is from a FIB sample with step-like thickness variation to begin with. The left part is much thinner than the right part. Therefore, we only see the pores in the thinner area. We have added relevant discussion in the revised manuscript.

Figure R5 Low magnification STEM image of cross-sectional sample with thin and thick area at RT and 400 °C.

Reviewer #2 (Remarks to the Author):

I have gone through the whole rebuttal letter and I feel the authors have made satisfactory replies to all of my concerns and other reviewers. There are new experimental and theoretical results added, which substantially improved the manuscript. Reconstructions of topological materials are a topic that raising increasing

interest. Therefore I would like to recommend its publication in Nature Communications.

I have gone through the review from reviewer #3. For my personal opinion, the authors have already presented a systematic study elaborating the experimental observation and also its theoretical interpretation. The data are consistent, the experiments are nicely executed. In the last round of review, the authors have provided additional data both experimentally and theoretically, thus I think the story is quite complete. Though this is only done in the GBT material system, it has already demonstrated a unique case of structural reconstruction induced by ion diffusion. On this basis, I think the comment #1 and #3 are a bit picky to the authors, though #2 is reasonable since the authors can elaborate more on the impact of their studies.

Reply: We would like to thank the reviewer for the positive comments on our revised manuscript.

Reviewer #1 (Remarks to the Author):

Thanks for the satisfactory replies from the authors, all my concerns are resolved. I would like to suggest the acceptance of this paper.

Responses to Comments of Nature Communications NCOMMS-22-21516B

We would like thank the reviewers for their positive comments and useful suggestions.

REVIEWER COMMENTS

Reviewer #1 (Remarks to the Author):

Thanks for the satisfactory replies from the authors, all my concerns are resolved. I would like to suggest the acceptance of this paper.

Reply: We would like to thank the reviewer for the positive comments on our revised manuscript.